# CpG dinucleotide enrichment in the influenza A virus genome as a live attenuated vaccine development strategy

**Colin P. Sharp[1], Beth H. Thompson[1], Tessa J. Nash[1], Ola Diebold[1], Rute M. Pinto[1], Luke Thorley[1], Yao-Tang Lin[1], Samantha Sives[1], Helen Wise[2], Sara Clohisey Hendry[1], Finn Grey[1], Lonneke Vervelde[1], Peter Simmonds[3], Paul Digard[1], Eleanor R. Gaunt[1] \***

1 The Roslin Institute, The University of Edinburgh, Easter Bush Campus, Midlothian, United Kingdom, 2 Royal Infirmary of Edinburgh, NHS Lothian, Edinburgh, United Kingdom, 3 Nuffield Department of Medicine, University of Oxford, South Parks Road, Oxford, United Kingdom

\* elly.gaunt@ed.ac.uk

## Abstract

Synonymous recoding of RNA virus genomes is a promising approach for generating attenuated viruses to use as vaccines. Problematically, recoding typically hinders virus growth, but this may be rectified using CpG dinucleotide enrichment. CpGs are recognised by cellular zinc-finger antiviral protein (ZAP), and so in principle, removing ZAP sensing from a virus propagation system will reverse attenuation of a CpG-enriched virus, enabling high titre yield of a vaccine virus. We tested this using a vaccine strain of influenza A virus (IAV) engineered for increased CpG content in genome segment 1. Virus attenuation was mediated by the short isoform of ZAP, correlated with the number of CpGs added, and was enacted via turnover of viral transcripts. The CpG-enriched virus was strongly attenuated in mice, yet conveyed protection from a potentially lethal challenge dose of wildtype virus. Importantly for vaccine development, CpG-enriched viruses were genetically stable during serial passage. Unexpectedly, in both MDCK cells and embryonated hens' eggs that are used to propagate live attenuated influenza vaccines, the ZAP-sensitive virus was fully replication competent. Thus, ZAP-sensitive CpG enriched viruses that are defective in human systems can yield high titre in vaccine propagation systems, providing a realistic, economically viable platform to augment existing live attenuated vaccines.

## Author summary

CpG dinucleotides are under-represented in vertebrate genomes, wherein cytosines in the CpG conformation are methylated to regulate transcription. Methylated cytosines are prone to deamination, resulting in TpG dinucleotides replacing CpGs. The resultant CpG suppression has provided a route by which vertebrate cells can recognise RNA from invading pathogens, using cellular Zinc-finger Antiviral Protein (ZAP) as a CpG sensor. Vertebrate-infecting RNA viruses also genomically suppress CpGs, and it is believed that this is an evolved trait to evade detection by ZAP. Here, we engineered an influenza A

**Data Availability Statement:** The authors confirm that all data underlying the findings are fully available without restriction. All relevant data are

within the paper and its Supporting Information files.

**Funding:** FG, LV, PD and EG are supported by a BBSRC Institute Strategic Programme grant (BB/P013740/1). BT was supported by a Microbiology Society Harry Smith Vacation studentship (GA002550) and a Carnegie Trust Undergraduate Vacation Scholarship (VAC011866). FG and PD are supported by BBSRC grant no BB/S00114X/1. OD is supported by a Roslin Studentship Award. LT is supported by an EASTBio Studentship Award. PS is supported by Wellcome Investigator Award (WT103767MA). LV and PD are also supported by the European Union's Horizon 2020 research and innovation programme under grant agreement no. 727922 (DELTA-FLU) and SS under grant agreement no. 731014 (VetBioNet). TN was supported by an iCase studentship from the BBSRC (BB/MO14819). EG is also supported by a Wellcome Trust/ Royal Society Sir Henry Dale Fellowship (211222_Z_18_Z). The funders had no role in study design, data collection and analysis, decision to publish, or preparation of the manuscript.

**Competing interests:** The authors have declared that no competing interests exist.

virus (IAV) with elevated CpG content and characterised how this impacts viral replication. CpG addition resulted in viral attenuation, mediated by ZAP activity. CpG suppression is conserved in dog and chicken genomes (relevant for live attenuated IAV vaccine propagation), and it is logical to predict that ZAP-mediated CpG sensing would also be conserved in these species. However, when we propagated ZAP-sensitive IAV in cognate culture systems, we saw no replication defect. This unexpected result raises questions about why viruses infecting these species suppress CpG in their genomes, and importantly delivers a new, tractable approach to augment rational live attenuated IAV vaccine design.

## Introduction

Single stranded RNA (ssRNA) viruses typically under-represent CpG dinucleotides in their genomes [1–3]. CpG dinucleotides in viral transcripts may be bound by zinc-finger antiviral protein (ZAP) [4], which causes an antiviral state in the infected cell. This provides a basis for observations from several studies showing that engineering viral genomes enriched for CpG yields replication-defective viruses (reviewed in [5]).

Four ZAP isoforms are currently identified, including two major isoforms, ZAPS (short) and ZAPL (long), produced by alternative splicing [6]. All isoforms incorporate a common N-terminal region that contains four zinc-finger motifs with RNA-binding function [7–10]. CpG-mediated antiviral activity has been demonstrated for both ZAPS [4], and for ZAPL [11]. ZAPS is an interferon-stimulated gene (ISG) [9,12–14], whereas ZAPL is constitutively expressed [6]. For its antiviral function, ZAPS must interact with its binding partner, the RNA-binding E3 ubiquitin ligase-like tripartite motif containing protein 25 (TRIM25) [15,16]. The current understanding is that ZAPS binds CpG motifs on viral RNA, and upon subsequent binding by TRIM25 and catalytic activation, a downstream cascade leads to the induction of type I and type II interferons (IFN) [12,17]. ZAPS may also directly bind RNA leading to exonuclease-mediated degradation [18], and so the antiviral activity of ZAPS appears multifaceted (reviewed in [5]). A newly discovered ZAPS binding partner, KHNYN, has so far only been found to be required for ZAP-mediated antiviral activity during lentivirus infections [19,20]. While ZAPS has so far been more extensively studied, interactions of ZAPL with TRIM25 and KHNYN have been shown to be more efficient than those of ZAPS [11]. CpG-specific binding by ZAP is likely an evolutionarily conserved trait across tetrapods that is not retained in avian species [21]; nevertheless, chickens express ZAPL but not the shorter isoform [22].

Due to the replication defects imparted by CpG introduction into viral genomes, CpG enrichment has widely been proposed as a potential strategy for the development of live attenuated vaccines. However, if viral growth is impaired, then the large-scale virus propagation required for vaccine manufacturing [23] is not viable. The identification of ZAP as a CpG sensor [4] delivered the possibility that CpG-enriched viruses could be grown in ZAP knockout (KO) systems, thus restoring viral titre in an artificial system, with the replication defect maintained in a vaccine recipient [5]. This principle has been validated using enterovirus-71 as an exemplar [24], although there is currently no licensed vaccine targeting this virus.

Live attenuated vaccines for influenza A viruses (IAV) are used in children in the USA and Europe [25,26], with up to 14 million doses delivered annually [27]. IAV vaccines represent an ideal target for augmentation by incorporating additional CpGs, as this approach has the potential to overcome some of the significant challenges arising during vaccine manufacture. IAV live vaccines are propagated in either embryonated hens' eggs or Madin-Darby canine

kidney (MDCK) cells, and both IAV vaccine propagation systems can be subjected to genome editing [28,29] to improve virus yield. For egg based vaccines, unpredictable supply and ethical challenges mean that there is a consensus about the need to move towards cell culture generated vaccines [30,31]. However, MDCK based vaccines are more expensive and large-scale manufacture will require establishment of new production pipelines [32,33]; therefore, this transition is not straightforward and egg-based vaccines will be commercialised for the foreseeable future.

Recoding of the IAV genome (i.e., synonymous alterations of the genetic sequence) is achieved using the facile reverse genetics system [34,35], in which the 8 segments of negative sense RNA genome are encoded on plasmids that upon co-transfection into permissive cells yield infectious virus. The IAV genome has been thoroughly characterised, with well-defined packaging signals [36–38] and gene annotation (including accessory proteins, reviewed in [39]), providing sufficient clarity about which genome regions may be suitable for recoding and those that must remain unaltered.

To determine if CpG enrichment can enhance live attenuated vaccines, we used an IAV vaccine strain to test the effects of adding CpGs to segment 1, which encodes the polymerase protein PB2 and the accessory protein PB2-S1. This CpG enrichment resulted in a virus that was attenuated in human cell systems, with wildtype fitness restored by ZAP KO. The CpG-high virus was genetically stable in ZAP-positive cells, and showed no growth defects in embryonated hens' eggs or MDCK cells. Therefore, we have provided a pathway to develop CpG-enriched live attenuated IAV that can be propagated to high yields using standard approaches.

## Results

### Addition of CpG dinucleotides to the IAV genome causes ZAPS-dependent replication defects

As ZAP recognises CpG-enriched transcripts [4] and the IAV genome naturally suppresses CpG, we investigated whether adding CpGs to the IAV genome resulted in viral replication deficiencies and whether attenuation could be offset in ZAP -/- cells. We engineered a CpG-high ('CpGH') PR8 (A/Puerto Rico/8/1934 H1N1) virus through synonymous addition of 126 CpGs into segment 1, including compensatory mutations to maintain individual base frequencies across the recoded regions. Additionally, the same viral genomic region was synonymously recoded by codon rearrangement to create a control ('CDLR') virus. This allows the detection of any attenuation caused by disruption of e.g. RNA structures or alternative open reading frames. The panel of recoded viruses is summarised (**Table 1**).

The CpGH IAV displayed a significant ~2-$\log_{10}$ defect in production of infectious virus progeny in A549 cells, whereas the CDLR virus was fully replication competent (**Fig 1A**). In paired ZAP -/- cells (a kind gift from the laboratory of Prof. Sam Wilson, MRC-University of Glasgow Centre for Virus Research, UK; validated in **S1A Fig**) [40], the CpGH virus replicated as well as wild type (WT) PR8 (**Fig 1A**), suggesting that the replication defect in the CpGH virus is ZAP dependent.

A construct that was designed to be attenuated by the introduction of UpA dinucleotides added across the same region of segment 1 (UpAH) was also defective in A549 cells, but in contrast with the CpGH virus, did not recover fitness in ZAP-depleted cells (**Fig 1A**). This illustrates the specificity of ZAP for alleviating CpG-mediated attenuation, and is in contrast with paralogous experiments using echovirus-7, where an attenuated UpAH virus had fitness restored when ZAP was depleted [41].

**Table 1. Summary of the genetic properties of CpG modified influenza A viruses that were recoded in segment 1.**

| Virus backbone | Mutation | No. CpGs | Δ CpG | No. UpAs | Δ UpA | CAI |
|---|---|---|---|---|---|---|
| A/Puerto Rico/8/1934 | Wildtype (segment 1) | 51 | - | 108 | - | 0.699 |
| A/Puerto Rico/8/1934 | CDLR | 52 | +1 | 108 | - | 0.696 |
| A/Puerto Rico/8/1934 | CpGH | 177 | +126 (347%) | 103 | -5 | 0.625 |
| A/Puerto Rico/8/1934 | 5'-CpGH | 131 | +80 (257%) | 107 | -1 | 0.653 |
| A/Puerto Rico/8/1934 | 3'-CpGH | 97 | +46 (187%) | 104 | -4 | 0.668 |
| A/Puerto Rico/8/1934 | CpG_M | 97 | +46 (187%) | 103 | -5 | 0.680 |
| A/Puerto Rico/8/1934 | UpAH | 47 | -4 | 223 | +115 | 0.661 |

CAI, codon adaptation index. CAI scores have been calculated relative to the human genome (A/Puerto Rico/8/1934) in https://www.bioinformatics.nl/emboss-explorer/output/806185/ compared to the human codon usage profile. For full construct sequences, please refer to **S1 Table**.

ZAP binding partner TRIM25 is required for optimal ZAP antiviral activity [15,16] and so replication of WT and engineered viruses was measured in TRIM25-/- cells (a kind gift from the laboratory of Prof. Gracjan Michlewski, International Institute of Molecular and Cell Biology, Warsaw, Poland; validated in **S1B Fig**) [42]. In WT HEK293 cells, CpGH IAV was again significantly replication-impaired. As with ZAP KO, TRIM25 KO also resulted in rescued replication of CpGH virus (**Fig 1B**).

To determine whether the observed phenotypes could be recapitulated in primary cells, siRNA knockdown of ZAP or TRIM25 was applied in Normal Human Dermal Fibroblasts (NHDF; primary human foreskin fibroblasts) as previously described [43] (**Fig 1C**). While ZAP knockdown recapitulated rescue of CpGH virus fitness, TRIM25 knockdown did not (**Fig 1D**). TRIM25 knockdown was incomplete (Fig 1C), and so this may indicate that sub-physiological levels of TRIM25 are sufficient for its role in CpG recognition, or that ZAP-TRIM25 interactions are less important in these NHDF cells.

KHNYN is a recently identified ZAP-binding partner [19] and so its impact on replication of the virus panel was also tested (cells were a kind gift from the laboratory of Dr Chad Swanson, King's College London, UK; **S1C Fig**). As in Fig 1A, a significant defect in the replication of CpGH virus was seen in WT A549 cells. In paired KHNYN-/- cells, CpGH virus titres remained significantly lower than those of WT PR8. However, the fold difference in titre between WT PR8 and CpGH was significantly reduced, indicating a partial rescue of CpGH virus replication in the absence of KHNYN and suggesting that KHNYN is required for optimal CpG sensing (**Fig 1E**).

To determine whether we could reduce the number of CpGs in the CpGH construct and maintain the same level of replication inhibition, the recoded region of segment 1 was split so that either only the 5' two thirds of recoding were present (as viewed in positive polarity; '5'-CpGH'), or the 3' third ('3'-CpGH'; **S1D and S1E Fig**). As an alternative, CpGs were introduced across the entire recoded region at a lower density than CpGH but at the same frequency as 3'-CpGH ('CpG-M'), to test whether CpG frequency or density were of primary importance. This virus panel was used to infect A549 cells. As previously, the CpGH virus was significantly attenuated. However, reduction in CpG frequency or density diminished this attenuation, with a dose-dependent effect observed (**Fig 1F**). No attenuation was apparent for either 3'-CpGH or CpG-M, which had the same number of CpGs introduced at high and low density respectively. This is in stark contrast with a CpG-enriched HIV-1 replicon, in which single CpGs contributed to viral attenuation, and the distance between CpG sites was critical [24]. Our CpG-M construct incorporated equivalent CpG density to the HIV-1 replicon but was not significantly attenuated, highlighting that the importance of CpG number and spacing

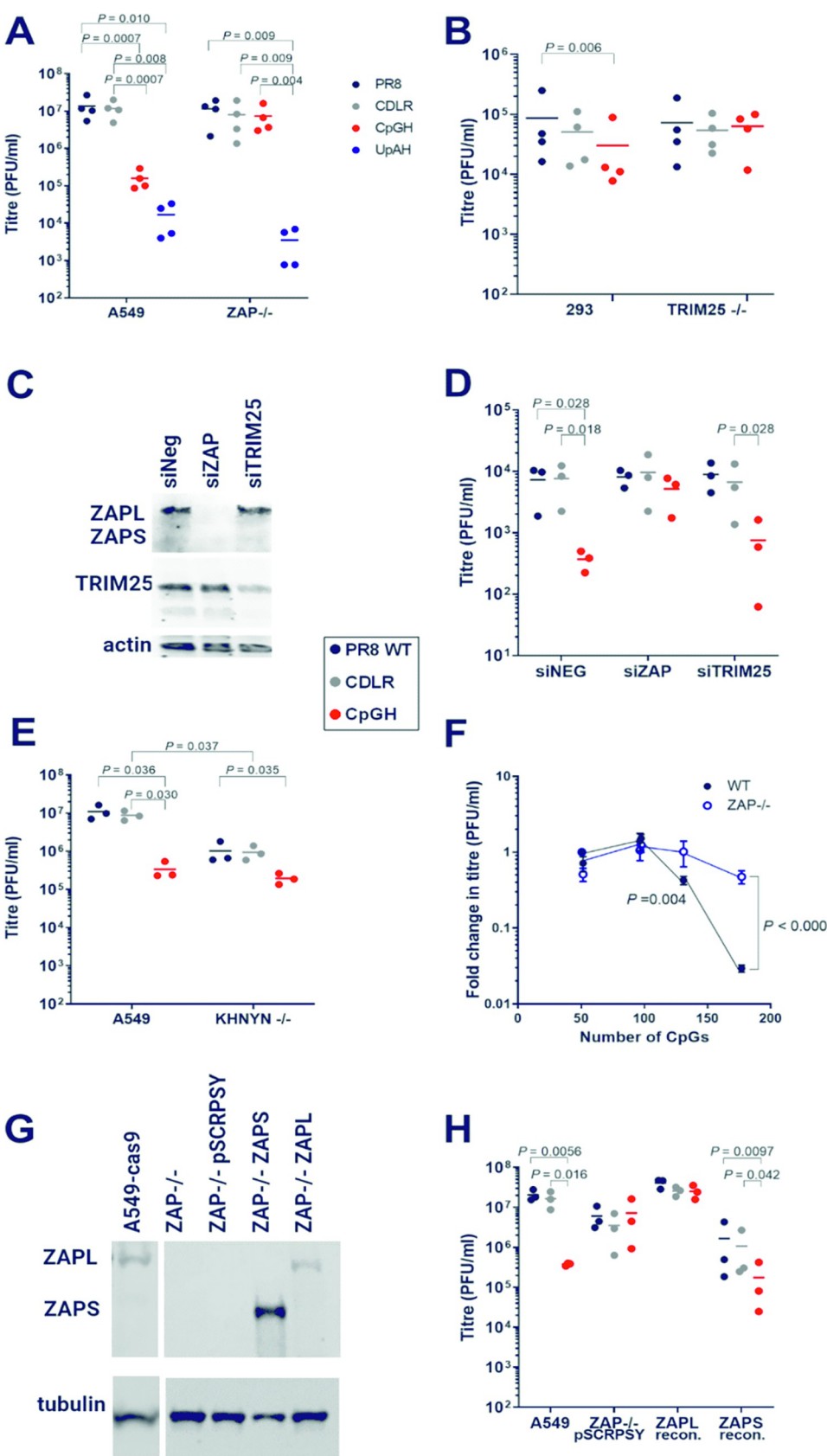

**Fig 1. Influenza A viruses with CpG enrichment in segment 1 have replication defects mediated by ZAP.** WT PR8, synonymously permuted control virus (CDLR) and CpG enriched (CpGH) viruses were used to infect various cell types at low MOI for 48 hours. Synonymous recoding was performed in segment 1. Viruses were grown in A549 and paired ZAP -/- cells (**A**) or in 293 and paired TRIM25 -/- cells (**B**). **C.** siRNA knockdown of ZAP or TRIM25 in NHDF primary cells was confirmed by western blot. **D.** After knockdown, NHDFs were infected at high MOI for 24 hours. Impact of ZAP or TRIM25 knockdown on replication of the seg1 virus panel in NHDFs was determined. **E.** Seg1 virus panel was grown in A549 and paired KHNYN -/- cells. **F.** The number of CpGs introduced into the CpGH construct was depleted by a third (80 CpGs added, giving a segment total of 131) or by two thirds (46 CpGs added, giving a segment total of 97). When 46 CpGs were added, this was done either by retaining the 46 CpGs added to the 3' end of RNA (positive orientation), or by spacing the 46 CpGs added across the segment. This panel of CpG mutants, along with CDLR and CpGH, were titred in WT A549s and paired ZAP-/- cells as in (A). **G.** ZAPL/ ZAPS expression was reconstituted in A549 cells using lentivirus expression constructs, and isoform-specific expression was confirmed by western blotting. **H.** A549 ZAP-/- cells were reconstituted for isoform-specific ZAPL or ZAPS expression, and titres of seg1 virus panel were determined. For validation of KO phenotypes, please refer to **S1 Fig**. Confocal visualisation of ZAP reconstitution is presented in **S2 Fig**.

varies between systems. Codon pair and codon usage distributions of the virus panel are summarised (**S1F and S1G Fig**).

To determine whether ZAP sensitivity of the CpGH virus was mediated by a specific ZAP isoform, expression of ZAPL or ZAPS was reconstituted in A549 ZAP -/- cells using pSCRPSY lentiviral vectors (**Fig 1G** and **S2 Fig**). In ZAP-/- cells transduced with empty pSCRPSY no differences in replication were observed between WT PR8, CDLR and CpGH viruses, indicating that the lentivirus vector did not impair CpGH virus replication (**Fig 1H**). When ZAPL expression was reconstituted, all viruses showed a modest increase in replication, with CpGH virus displaying similar titres to those of WT PR8. It is possible that global increases in viral replication are attributable to the importance of ZAPL for maintenance of cell viability under stress conditions, such as viral infection [44]. However, in cells overexpressing ZAPS, all viruses in the panel showed a replication defect, in keeping with its antiviral properties. Nevertheless, CpGH virus titres were significantly lower than WT PR8, indicating that ZAPS is the CpG sensor for CpGH IAV.

## CpG enrichment does not impair IAV virion assembly

The mutations introduced into the PR8 genome were designed to avoid classically described packaging signals present at the termini of viral genome segments [36,45]. However, recent work has highlighted the potential for the middle regions of different segments to interact during virion assembly [38]. Therefore, we questioned if the mutations made in segment 1 disrupted viral genome packaging. Firstly, genome copy:PFU ratios were determined for the panel of virus stocks. RNA copy numbers for both segment 1 and segment 5 were analysed. A previously characterised packaging-defective mutant, '4c6c' [36,46] was included as a control. Relative to WT PR8, no significant impact on RNA:PFU ratios was observed for either the CDLR or CpGH viruses (**Fig 2A**).

Secondly, electrophoretic separation of RNA extracted from purified virus particles was performed, to visually determine whether synonymous recoding altered the amount of RNA packaged for each segment. For each virus, equal numbers of infectious particles were pelleted through sucrose, then RNA was extracted from viral pellets and separated by urea-PAGE. As expected, the PR8 WT virus yielded 7 bands corresponding with 8 segments of RNA (segments 1 and 2 co-migrate as they are the same size; **Fig 2B**). The CDLR control band densities were visually similar to those of the WT virus. For CpGH virus, while it did not appear that there were any differences in band densities (in keeping with the genome copy:PFU ratios), an apparent retardation in the electrophoretic mobility of CpG-enriched segment 1 was observed.

As the CpGH construct was compositionally corrected so that all bases are equally represented, the electric charges should be identical for all the segment 1 RNAs. RNAs were run on

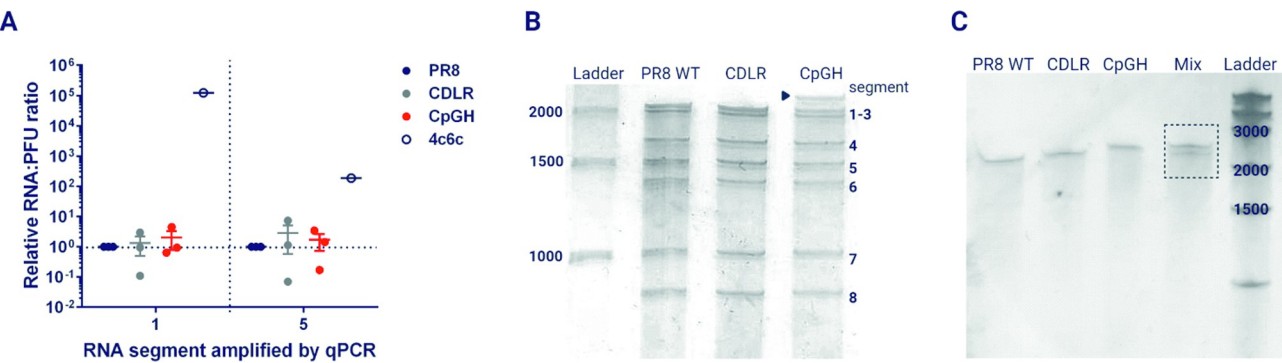

**Fig 2. CpG enrichment does not affect packaging but may alter electrophoretic mobility. A.** Virus stocks of the seg1 virus panel were titred by plaque assay, and then the relative copy numbers of segment 1 and segment 5 were determined by qPCR. RNA:PFU ratios were then calculated. A '4c6c' mutant virus [36,46] was used as a packaging mutant control. To visualise packaged viral RNAs, viral stocks were purified by ultracentrifugation and subjected to urea-PAGE, thereby separating the 8 segments of the viral genome (**B**). A shift in the electrophoretic mobility of segment 1 of the CpGH virus was observed (large black arrow), and in order to determine whether this was due to RNA modification, RNAs generated through *in vitro* transcription (i.e. a cell-free system, free from RNA modifying enzymes) were run under the same conditions (**C**). For 'mix' samples, equimolar combinations of CDLR and CpGH constructs were combined and run to demonstrate whether a mobility shift was apparent in CpGH constructs.

denaturing gels, and so RNA structure should also not influence mobility. To determine whether a post-transcriptional modification might explain the diminished electrophoretic mobility of the CpG-enriched RNA, segment 1 transcripts were synthesised by *in vitro* transcription, so that no cellular-derived post-transcriptional modifications could occur. When equimolar CDLR and CpGH constructs were mixed together and subjected to electrophoretic separation, their different migration phenotypes was maintained (**Fig 2C**). Most likely, CpG introduction has altered secondary structures in the CpG-enriched RNA transcripts that survived denaturation and retarded electrophoretic migration. Nevertheless, we saw no evidence that increasing CpG frequencies deleteriously affected vRNA packaging.

## CpG enrichment does not impact m⁵C methylation status of IAV vRNAs

Methylation of cytosines in CpG dinucleotides of DNA regulates transcription [47], and cytosine methylation of RNAs has been described [48,49]. We therefore hypothesised that adding CpG sites to RNA may increase the frequency of cytosine-associated methylation. Viral genomic RNA was purified from virions propagated in embryonated hens' eggs, and bisulphite conversion reactions, in which cytosines in purified RNA are converted to uracil unless protected from conversion by a methyl group, were performed. Converted RNAs were then sequenced and compared to non-converted sequences to identify potential methylation sites. A strongly methylated site at position 3256 of chicken 28S ribosomal RNA (GenBank accession # XR_003078040.1) served as a positive control. Methylation frequencies of 97.0–98.8% were detected at this site across all samples, indicating that the protocol worked efficiently (**Fig 3A**).

Deep sequencing reads of the bisulphite converted RNAs were mapped to the *in silico* converted reference sequences, which yielded maximal methylation signals of ~13% across all nucleotide positions (**Fig 3B–3D**; PR8, CDLR and CpGH respectively). However, in WT PR8, three obvious peaks in methylation signal were identified, each occurring over regions of ~50 bases, that were evident in up to approximately 10% of sequences (**Fig 3B**). The first of these clusters was identified in a region that was not mutated in CDLR or CpGH viruses and the signal was maintained across all three viruses, indicating a biologically reproducible phenomenon. The second two clusters at nucleotide positions ~800 and ~1400 were not retained in the mutant viruses that had been recoded in these regions (**Fig 3C and 3D**), indicating that

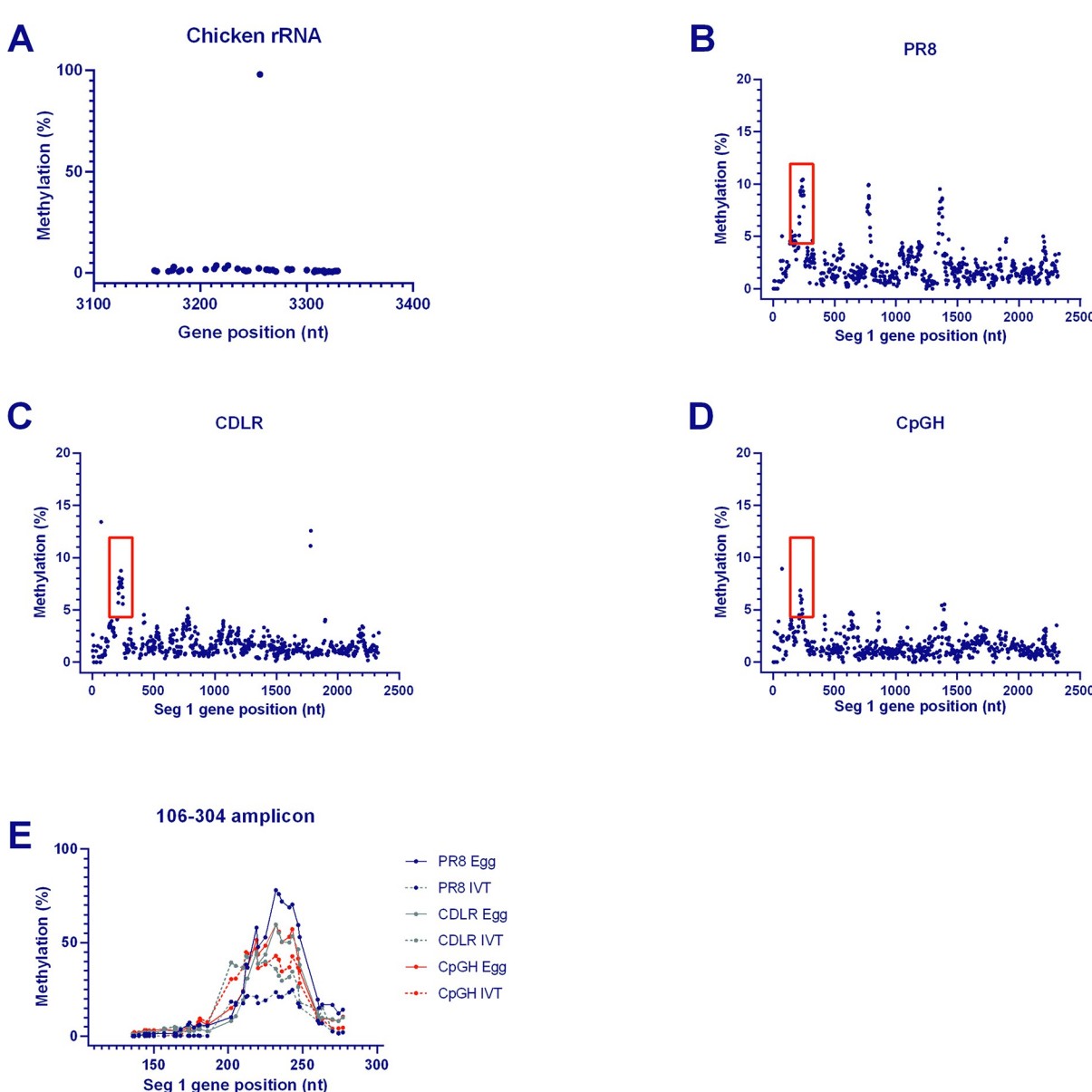

**Fig 3. CpG enrichment of viral RNA does not alter the methylation profile.** Segment 1 modified viruses were processed through bisulphite conversion reactions and then deep sequenced to determine the frequency of methylation in unmodified versus CpG-enriched transcripts. **A.** As virus reconstitution was performed in embryonated hens' eggs, chicken ribosomal RNA was sequenced from viral stocks as a positive control due to a known methylation site in this RNA. **B.** 5-methylcytosine signal intensity across PR8 segment 1 vRNA. **C.** 5-methylcytosine signal intensity across CDLR segment 1 vRNA. **D.** 5-methylcytosine signal intensity across CDLR segment 1 vRNA. **E.** 5-methylcytosine signal intensity across PCR amplicons of either vRNA (PR8 egg/ CDLR egg/ CpGH egg) or *in vitro* transcribed (IVT) RNA, across segment 1 positions 106–304. This region was selected due to the spike in signal observed across segment 1 vRNAs for the segment 1 virus panel (B-D, red boxes). For PR8 whole genome methylation plots please refer to **S4 Fig**.

mutation at these sites removed the methylation signals. Unlike the single base methylation in the chicken rRNA, these sites of apparent viral RNA methylation were centred around ~50 nucleotide regions with diminishing signal flanking the peak. This suggests that these clustered methylation signals may be false positives, possibly generated by RNA secondary structures that did not allow RNA unfolding, thus protecting cytosines from base conversion. To test whether the methylation signal was retained in RNAs synthesised in a cell-free system where

post-transcriptional modifications cannot occur, *in vitro* transcribed segment 1 RNAs for WT PR8, CDLR and CpGH were generated. These RNAs, along with paired vRNAs, were then subjected to bisulphite conversion, PCR amplification and sequencing across the target region. Methylation signals were retained in amplicons generated from the *in vitro* transcribed RNAs (**Fig 3E**), confirming that the methylation clusters were artefacts. The methylation signal was higher from this targeted sequencing, most likely because PCR amplification biased the sequencing output. Through whole genome sequencing of bisulphite converted RNA, we found no evidence of high efficiency methylation at any site in the PR8 genome (**S3 Fig**), and CpG enrichment did not introduce methylation sites.

## ZAP-mediated inhibition of CpG enriched viruses is independent of the type I IFN response

ZAPS has been reported to both stimulate [12] and inhibit [50] type I interferon responses, and so we tested whether the ZAPS-sensitive CpG-enriched IAV also induced IFN [51,52]. Supernatants from infected A549 cells were used in a HEK-Blue assay, which gives a colorimetric signal proportional to the levels of IFN-α/β present. As anticipated, by expressing a functional NS1 [53], WT PR8 did not induce detectable type I IFN (**Fig 4A**). In contrast, a control virus mutated to impair function (R38K41A; [54]) strongly induced IFN signalling. The CDLR control and CpGH viruses also did not induce type I IFN above background signal. This suggests that the attenuation of CpGH virus by ZAP is independent of any type I IFN-stimulating activity by ZAPS [12].

It was possible that IAV NS1 suppressed ZAPS activity sufficiently to block ZAP-mediated IFN induction by CpGH IAV. To test this, we introduced mutations into NS1 that have been described to specifically inhibit its anti-ZAP or anti-TRIM25 activity ([55,56]; **Table 2**). Introducing these NS1 mutations into the CpGH virus should alleviate NS1-dependent suppression of the ZAP/TRIM25 pathway, and so if NS1 does inhibit CpG sensing by ZAP, the defect in replication of the CpGH virus would be exacerbated by these mutations. However, this was not the case. In low MOI infections in A549 cells, viruses with mutations in NS1 at positions 95/99 (purportedly unblocking anti-TRIM25 activity [55]) all had similar titres to those of viruses without NS1 mutations (**Fig 4B**). For all viruses with mutations in NS1 at positions 96/97 (reportedly alleviating anti-ZAP activity), titres were down by 2–3 $\log_{10}$, indicating a general replication defect consistent with reports indicating these mutations have a deleterious effect on NS1 stability [55,57]. Viruses with mutations in NS1 at positions 107–110 (targeting anti-ZAP functionality) had similar titres to those of viruses without NS1 mutations. The replication defect of CpGH IAV relative to WT PR8 was consistent regardless of the NS1 mutations introduced, indicating a lack of specificity of these NS1 sites for ZAP/TRIM25 regulation.

Across the virus panel, CpGH viral protein accumulation (PB2 and NP) was lower than for counterpart WT viruses, reflecting the defective replication of this virus (**Fig 4C**). WT PR8, CDLR and CpGH viruses all induced ZAPS expression to similar levels.TRIM25 was also consistently expressed across samples.

Type I IFN induction by the segment 1-segment 8 mutant virus panel was examined (**Fig 4D**). For viruses with mutations introduced into NS1 amino acid residues 95 and 99, no significant type I IFN activity was detected in any of the segment 1-segment 8 virus panel. Conversely, when NS1 residues 96 and 97 were mutated, all viruses induced IFN to similar levels. NS1 mutations at positions 107–110 conveyed detectable IFN signal for the CpGH virus and also for the CDLR virus (though, curiously, not for PR8 WT virus). Therefore, while CpGH IAV is ZAP sensitive (Fig 1), there was no correlation of ZAP anti-CpG activity with the type I IFN pathway in A549 cells.

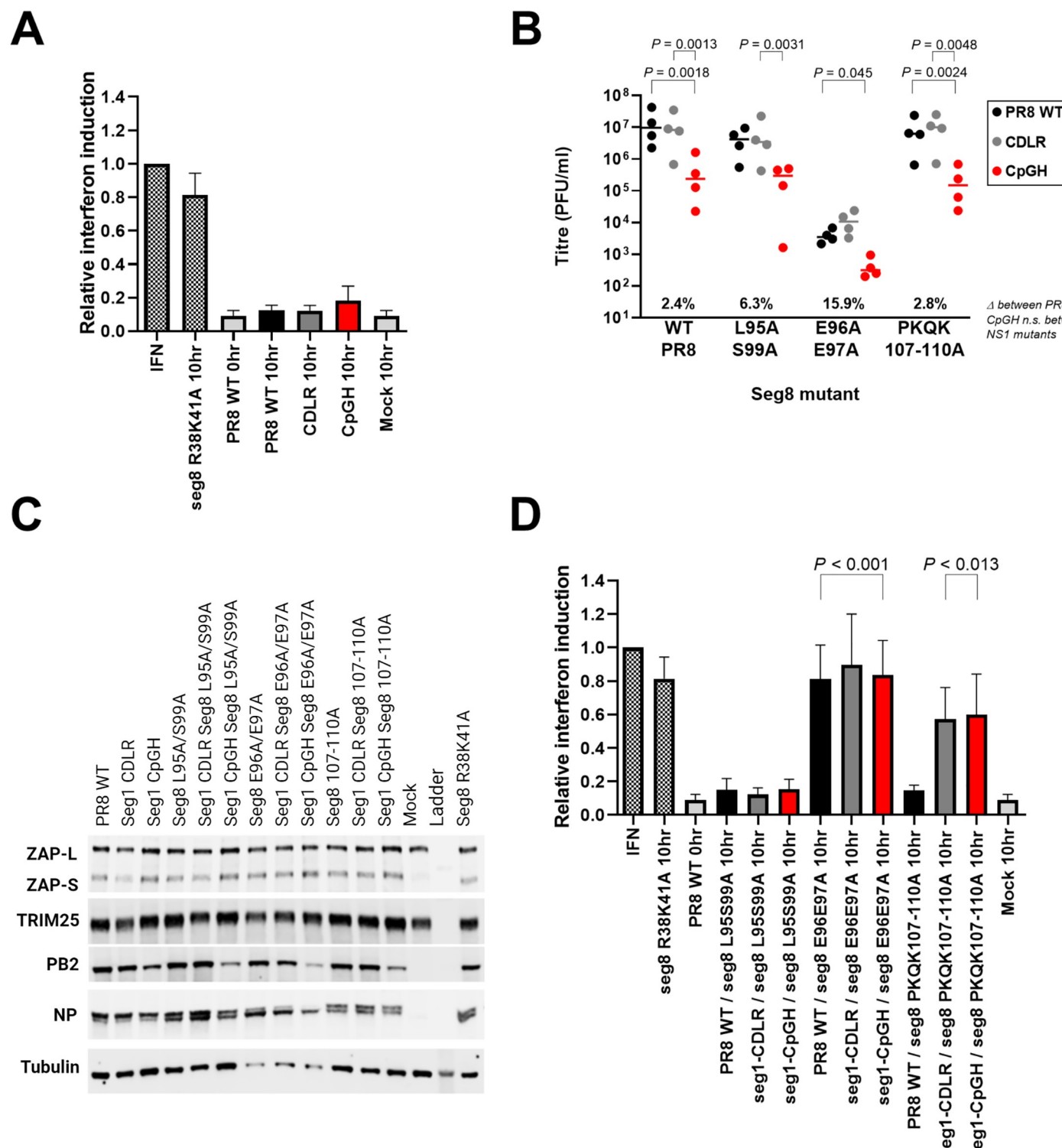

**Fig 4. Interaction of CpG-enriched viruses with the IFN pathway. A.** A549 cells were infected with WT PR8, CDLR or CpGH viruses at MOI 10 for 24 hours, and then type I IFN induction was quantified by HEK-Blue assay. Titrated IFN was used as a standard, and an IAV mutated in segment 8 to inhibit IFN induction (R38K41A) was used as a positive control. **B.** IAV WT PR8, CDLR or CpGH viruses were further mutated in segment 8 to abrogate the ZAP/TRIM25 targeting activity of NS1 encoded on this segment. Mutations introduced into segment 8 were L95S99A, reportedly defective in TRIM25 targeting [55], E96/97A, reportedly defective in ZAP targeting [56], and PKQK107-110A, reportedly defective in ZAP targeting [56]. Seg1/8 mutant viruses were used to infect A549 cells at MOI 0.01, for 48h, and viral titres were quantified. **C.** Western blots examining ZAP and TRIM25 expression in the same infections. **D.** The seg1/8 mutant virus panel was used to infect A549 cells at high MOI for 10 hours, and then IFN induction was quantified by HEK-Blue assay (A640 read-out).

**Table 2. Previously characterised mutations in NS1 that reverse ZAP/TRIM25-mediated IAV attenuation.**

| NS1 target | Abrogating mutation | Mechanism disrupted | Reference |
|---|---|---|---|
| TRIM25 | L95/S99A | RIG-I ubiquitination/ vRNP binding | [55] |
| ZAP | E96/E97A | Can no longer prevent TRIM25-mediated ubiquitination of ZAP | [56] |
| ZAP | PKQK107-110A | Can no longer prevent TRIM25-mediated ubiquitination of ZAP | [56] |
| Interferon | R38K41A | NS1 can no longer bind RNA | [54] |

## CpG enrichment leads to IAV transcript turnover

ZAP has also been linked with transcript turnover [58], and so we next examined the impact of ZAP on transcription and translation of viral proteins, and on transcript turnover. We first assessed whether viral transcript and/or protein levels were affected by CpG enrichment during virus replication. Time-course experiments were performed at high MOI with samples collected at 1, 2, 4, 6, 8 and 12 hpi for quantification of total viral RNAs and proteins. Compared to WT PR8 and paired CDLR virus, while lower transcript levels for the CpGH virus did not achieve statistical significance, reduced PB2 production was apparent (**Fig 5A–5C**). Reduced

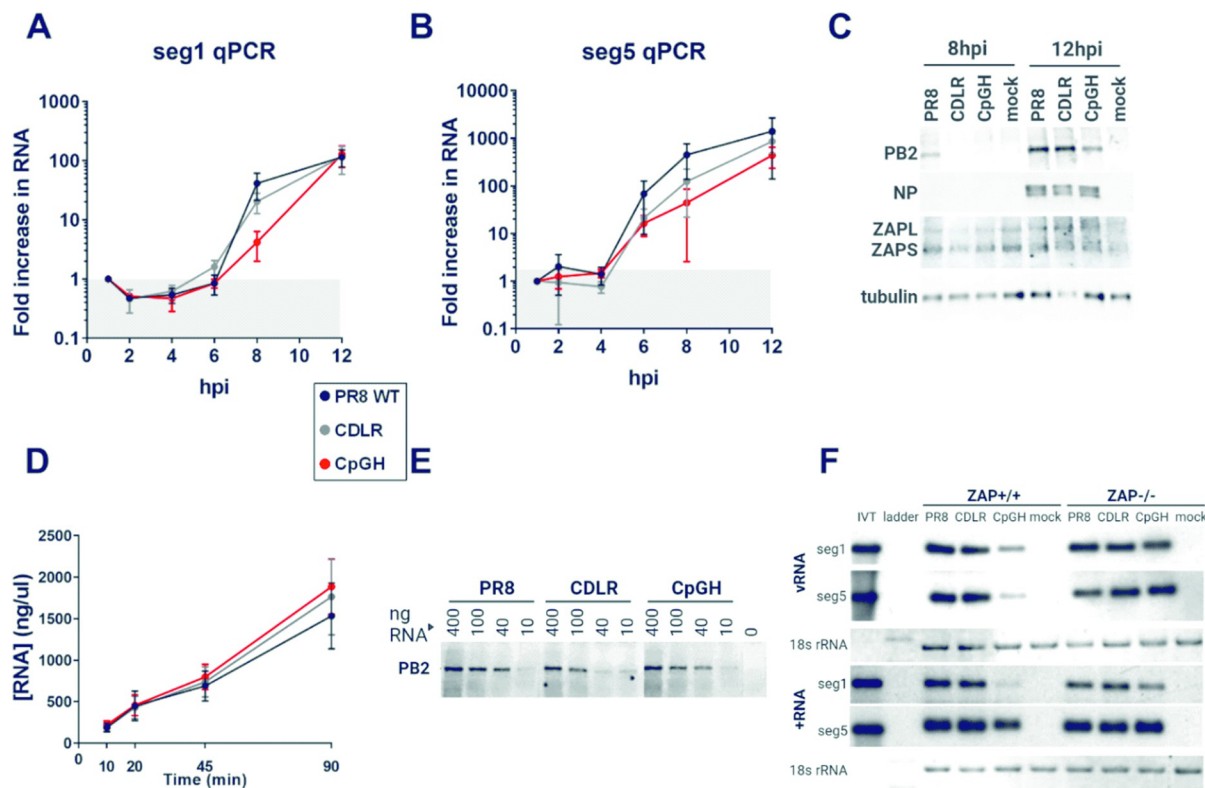

**Fig 5. CpG enrichment in the IAV genome results in reduced transcriptional levels during virus replication.** Transcription and translation were investigated in the context of infection using high MOI time courses. The virus panel was analysed for segment 1 (**A**) or segment 5 (**B**) RNA production. **C.** For assessment of protein production, PB2 (segment 1) and NP (segment 5) were assayed by western blot. **D.** Equal amounts of segment 1 WT (PR8), permuted control (CDLR) or CpG-enriched (CpGH) amplicons under T7 promoters were used in *in vitro* transcription assays to assess whether synonymous recoding impacted transcriptional efficiency in a cell free system. **E.** The efficiency of translation of RNAs was then assessed by titrating the synthesised RNAs in *in vitro* translation assays. **F.** A549 WT or ZAP-/- cells were infected with either WT PR8 or CpGH virus at an MOI of 15 for 8 hours, after which time RNA was harvested and probed by northern blotting using specific primers to detect segment 1 and segment 5 vRNA or positive polarity RNA. Segment 1 and segment 5 vRNAs or +RNAs were probed for on the same membrane, but due to differences in sensitives of the probes, different exposures are presented for the two segments.

polymerase activity as a result of CpG-enrichment was confirmed by minigenome assay, which reconstitutes the viral polymerase in cells in the absence of full infection (**S4 Fig**) [59]. As PB2 is a component of the polymerase complex, this did not distinguish between defective transcript and protein production.

To test whether segment 1 transcription and/or translation efficiencies were impaired in the absence of cellular machinery (e.g. ZAP), the two processes were decoupled and separately assayed in cell-free conditions. For *in vitro* transcription assays, equimolar amounts of segment 1 PR8, CDLR, and CpGH amplicons were used as input, and the amount of RNA produced was measured up to 90 minutes later. In the absence of cellular factors (and using the RNA-dependent RNA polymerase of bacteriophage rather than that of IAV), no differences in transcript production were seen across the panel (**Fig 5D**). Next, translational efficiencies were tested by titrating equimolar amounts of RNA in *in vitro* translation assays. No differences in protein production were apparent across the panel (**Fig 5E**). Thus, synonymous recoding imparted no inherent defect on transcription or translation in the absence of nucleic acid sensing pathways.

The qPCR data suggested a borderline reduction in CpGH virus RNA levels compared with those of the WT virus (Fig 5A and 5B), but may be confounded by the presence of partially degraded transcripts and by detection of RNAs of both polarities. To identify any reduction in full-length CpGH transcripts and differentiate between positive and negative polarity RNA species, northern blotting was performed. A549 and paired ZAP-/- cells were infected with PR8 WT, CDLR or CpGH viruses at MOI 10 for 8 hours, and RNA was isolated from cells. Positive and negative sense specific segment 1 and segment 5 probes were used for detection. Both positive and negative sense transcript levels were strongly reduced in WT A549 cells for the CpGH virus, whereas transcript levels were equivalent with those from the WT and CDLR viruses in ZAP-/ cells (**Fig 5F**). The most striking reduction was seen for segment 1 positive sense transcripts, which were diminished to the limit of detection. This profound reduction may have been somewhat masked in the qPCR data due to the presence of vRNAs and partially degraded segment 1 transcripts. In contrast, segment 5 positive sense transcripts–although of reduced intensity–were readily detectable. Taken together, this is indicative of a CpG-specific reduction in transcript abundance, supporting the conclusion that ZAP is targeting CpGH transcripts for degradation.

## CpG enrichment in IAV segment 1 causes viral attenuation in mice and attenuated CpGH IAV delivers protection from lethal dose challenge infection with WT virus

To test whether CpGH IAV is attenuated *in vivo*, groups of six, 6-week old female BALB/c mice were infected with 200 PFU (high dose) of the virus panel. All virus-infected animals lost weight after day 2 post infection, but mice infected with CpGH lost significantly less weight than mice infected with WT PR8 (**Fig 6A**). CpGH viral loads in the lung were significantly lower than those of WT PR8 and CDLR viruses (**Fig 6B**). Therefore, CpG-driven attenuation *in vitro* is reproduced in an *in vivo* system.

Next, we sought to determine whether the CpGH virus could cause a mild or clinically inapparent infection while offering subsequent protection from severe disease outcome, similar to a live attenuated vaccine. Mice were inoculated with low doses of virus, then subsequently challenged with a lethal dose of WT virus to establish the protective effect. For low dose infections, mice were infected with 20 PFU of either WT PR8 (n = 6), CDLR (n = 10), or CpGH virus (n = 10), or were mock infected (n = 10). Clinical signs were monitored for 10 days, by which time all mice were fully recovered. Mice infected with PR8 WT and CDLR displayed moderate clinical signs and weight loss, whereas symptoms in mice infected with CpGH virus were mild or entirely absent (**Fig 6C**). After 20 days, sera were collected from tail

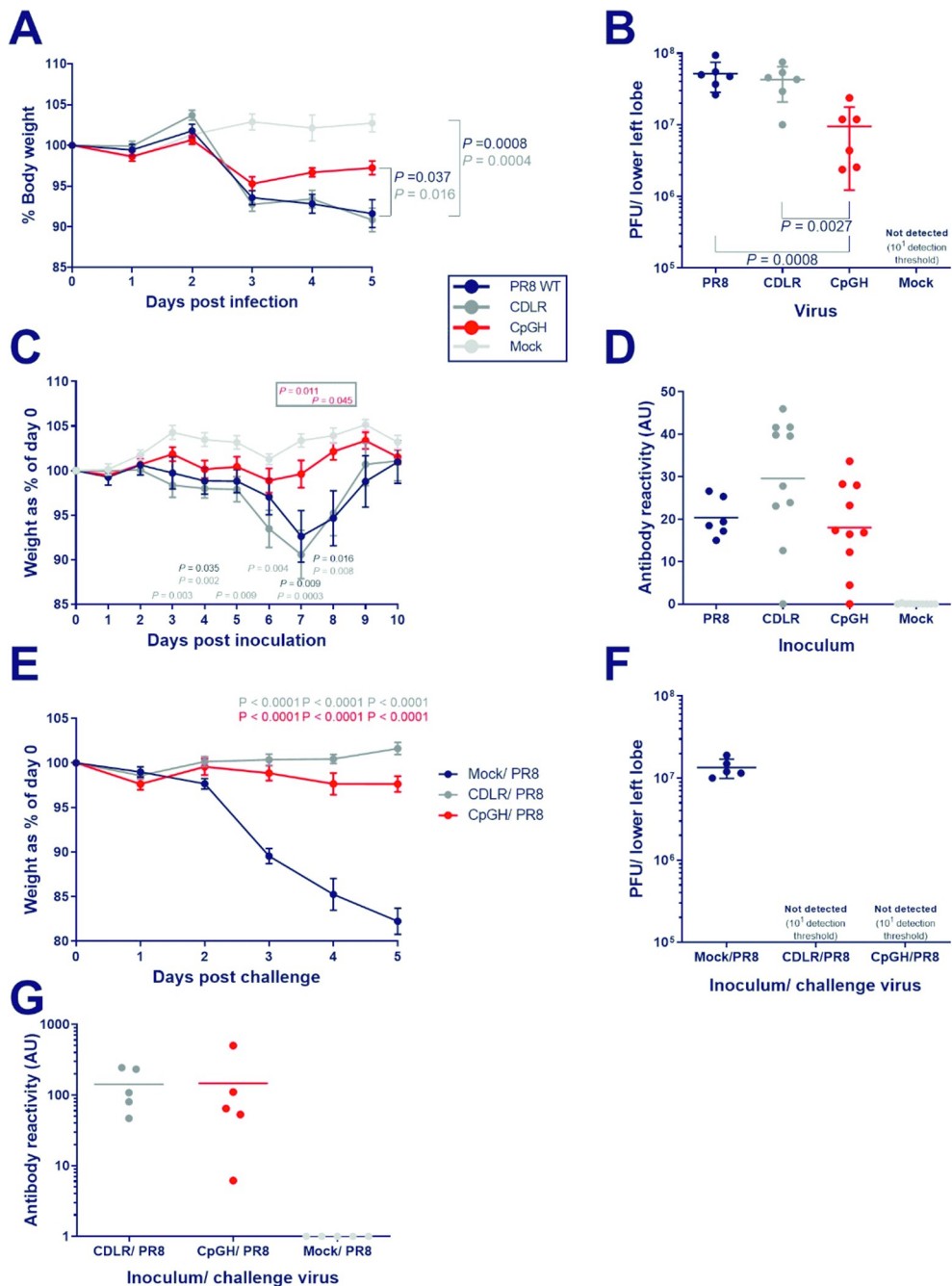

**Fig 6. CpGH IAV is attenuated in mice and conveys protection from challenge with a lethal dose of WT virus.**
Groups of six, 6 week-old female BALB/c mice were infected with 200 PFU of either WT PR8 virus, CDLR permuted control, CpGH virus or mock infected ($n = 2$) for 5 days; weight change measured daily (**A**). On day 5, lower left lung lobes were harvested and the presence of infectious virus was quantified (**B**). To determine the protective effect of CpGH virus infection, groups of 6–10 mice were infected with a low dose (20 PFU) of either PR8 (n = 6), CDLR control (n = 10), CpGH virus (n = 10) or were mock infected (n = 10) and weighed daily for 10 days until full recovery (**C**). At day 20 post-exposure, tail bleeds were used to collect sera for ELISA to quantify anti-IAV antibody levels (**D**). The next day, five mice exposed to either CDLR or CpGH virus, or mock exposed, were challenged with a potentially lethal dose of PR8 WT virus (200 PFU) and weighed daily for 5 days (**E**) at which time mice were culled and lower left lung lobes were collected for virus titration (**F**) and antibody levels in sera were quantified by ELISA (**G**). Sera were also tested for IAV protein binding using western blotting (**S5 Fig**).

bleeds for antibody semi-quantitation by ELISA. 6/6 PR8, 9/10 CDLR, 9/10 CpGH and 0/10 mock infected mice developed IAV-specific antibody responses; the levels of response to each virus varied but were not statistically different between groups (Fig 6D). After 21 days, the 6 PR8 mice, and 5/10 mice from the other infection groups were culled and sera were harvested for western blotting to confirm ELISA results (S5 Fig). Remaining mice were challenged with 200 PFU WT PR8 virus. After 5 days, this exposure was clinically inapparent in mice inoculated with the CDLR and CpGH viruses, but mice that were initially mock-inoculated developed severe clinical signs and significant weight loss (Fig 6E). These mice had high virus titres in the lung, whereas no virus was recovered from mice originally inoculated with CDLR or CpGH virus (Fig 6F). Mean antibody titres from these mice were 5-fold higher than levels at 20 days after exposure to the first virus dose, though again considerable variation was apparent (Fig 6G). Thus, the CpGH virus displayed the vaccine-desirable traits of causing mild or clinically inapparent infection, while offering full protection from challenge with a lethal virus dose.

### ZAP-dependent CpG sensitivity is absent in vaccine propagation systems

Live attenuated IAV vaccines are traditionally propagated in eggs, but there is an increasing impetus to move towards cell culture based propagation systems for ethical, medical and reliability reasons [33,60]. The impact of CpG enrichment on viral titres was therefore examined in both systems. Virus rescues were performed in embryonated hens' eggs, and titres obtained for the CpGH virus were reproducibly equivalent to those for WT PR8 (Fig 7A). Viruses were plaqued on MDCK cells; although the CpGH virus yielded slightly smaller plaques in these cells compared with PR8 WT (S6 Fig), titres from virus rescued in MDCK cells were equivalent across the virus panel (Fig 7A).

Possibly, a replication defect of the CpGH virus was not detected in eggs as titres had become saturated. To test this, eggs were inoculated at embryonic day 6 (ED6) and day 10 (ED10) with 100 PFU of PR8, CDLR or CpGH virus and harvested at 24 hours. Viruses replicated to higher levels in eggs at ED10, but no differences in titres between WT PR8 and CpGH virus was seen at either developmental time point (Fig 8B). Next, eggs at ED10 were inoculated with ten-fold serially diluted virus doses and embryonic survival was monitored for 4 days. Again, no differences were apparent between the WT PR8, CDLR and CpGH viruses, with all embryos dying at similar rates (S7 Fig). Nonlinear regression curves were fit for lethality across the dose range used (Fig 8C). LD50 values were within a five-fold range (PR8 = 0.29 PFU, CDLR = 1.27 PFU and CpGH = 0.55 PFU), thereby occurring within the limits of a single dilution factor for this experiment and so confirming no differences in embryonic lethality across the virus panel.

To determine whether the lack of fitness penalty for the CpGH virus was recapitulated in other chicken cells, chicken lung epithelial cells (CLEC213s) [61] were infected with the virus panel. Again virus titres were consistent across the panel (Fig 7D). As IAV is tropic for the gastrointestinal tract in birds [62], replication of the virus panel was also tested in chicken enteroids [63]. Again, CpGH virus replicated to similar titres as those of the WT virus (Fig 7E) and although viral protein production was variable, there was no evidence of a defect for the CpGH virus (Fig 7F). There was also no evidence of differential infection profiles either by confocal imaging (S8 Fig) or by qPCR array to analyse expression of 88 innate immune genes in chickens (S9 Fig).

Thus, CpGH IAV replicated equivalently to WT PR8 in the MDCK cells and in the embryonated hens' eggs used for vaccine virus propagation.

### Large scale recoding offers a genetically stable method of virus attenuation

Synonymous recoding of viral genomes as a method of attenuation for vaccine development is reliant on the modifications being genetically stable. To investigate the stability of CpG

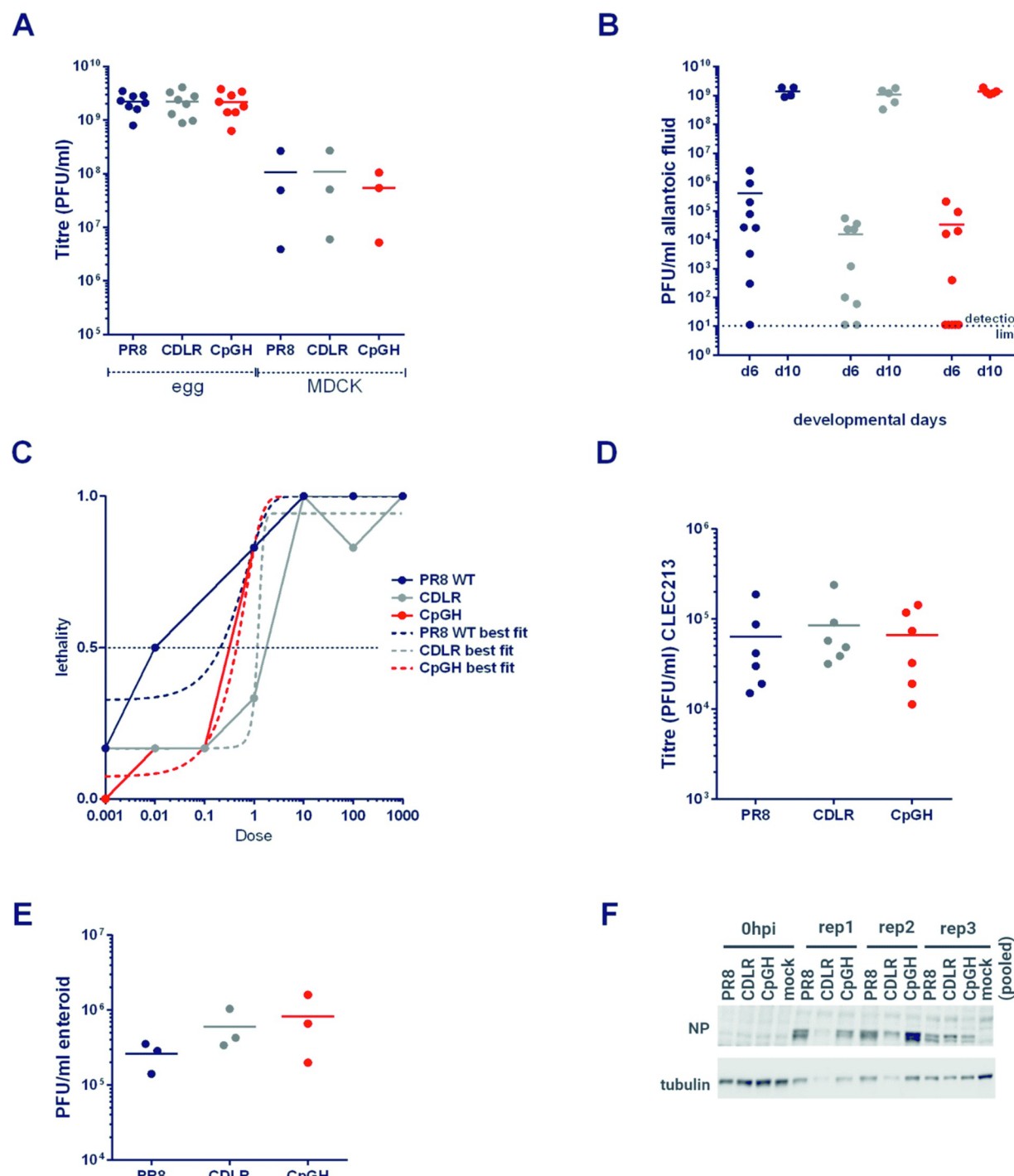

**Fig 7. A ZAP-sensitive attenuated CpG-enriched IAV replicates equivalently to WT virus in vaccine propagation systems. A.** PR8 WT, CDLR and CpGH viruses were grown in embryonated hens' eggs or in MDCK cells and titred. **B.** Embryonated hens' eggs were inoculated with 100 PFU virus at developmental day 5 or day 9, and viral titre in allantoic fluid was assessed 24 hours later. **C.** Lethality of the virus panel in eggs was fit to nonlinear regression curves for LD50 calculations. **D.** The virus panel was grown in chicken epithelial cell line CLEC213 and viral titres were quantified by plaque assay. **E.** The virus panel was used to infect chicken enteroids at an MOI of 10 for 24 hours, and the production of infectious virus, and viral protein production (**F**) were assayed. For confocal imaging of infected enteroids, please refer to S8 Fig. For innate gene expression analyses from infected enteroids, please refer to S9 Fig.

enrichment in the IAV genome, the virus panel was serially passaged ten times at low MOI in A549 cells. Passages were performed four times, with input virus for two experiments being derived from egg-grown stocks and two from MDCK-grown stocks. The ~1 $\log_{10}$ lower titre of CpGH virus compared with the WT or CDLR viruses was maintained over the course of the passage experiments, for both egg-derived (**Fig 8A**) and MDCK-derived (**Fig 8B**) virus stocks.

After the tenth passage, viruses were subjected to full-genome deep amplicon sequencing to analyse nucleotide changes occurring in their genomes. Sequencing coverage across the twelve viruses (4 replicates each of WT PR8, CDLR, and CpGH) ranged from 127–67,000 (**S10 Fig**).

To establish background signal from PCR and sequencing error, amplification and sequencing was performed on the PR8 plasmids used for virus rescue. In this control sample (**S10M Fig**), the highest mismatch versus database reference was at a single nucleotide position occurring in 2.3% of reads, and so this was used as the threshold above which mutations were considered real. The sum totals of mutations over this threshold arising in all egg- and MDCK-derived virus stocks after ten passages were calculated to include proportional representation for substitutions that had not reached fixation (for example, if a mutation was present in 10% of sequences, this was counted as 0.1 mutations). Consistent numbers of mutations were observed for CpGH virus compared with WT PR8 and control viruses, and so more mutations were detected per infectious virion for CpG enriched viruses (**Fig 8C**). However, the overall number of nucleotide changes identified in recoded viruses did not exceed those for WT PR8, indicating a genetic stability equivalent to WT virus. No difference in the mutation rates were observed between the egg-derived and MDCK-derived viruses. The numbers of mutations arising were broken down by segment (**Fig 8D**); the highest density of mutations occurred in segments 4 and 6 (encoding the viral surface proteins) and segment 8 (encoding NS1 and NEP), as previously described [64].

The recoded sites of the passaged CpGH virus were examined for evidence of mutation and reversion (**Fig 8E**). No nucleotide changes at altered sites became fixed in any of the four replicates. Sequence changes across the genomes of all serially passaged viruses are presented in **S11 Fig**. Overall, the data indicated that CpG enrichment is a genetically stable recoding strategy for IAV.

## Discussion

Synonymous recoding of viral genomes has been proposed as a potential strategy for live attenuated vaccine development [24,41,65]. However, such recoding inevitably impairs virus replication and so reduces vaccine yield. We have demonstrated that CpG enrichment is a viable vaccine enhancement strategy for IAV as we have made a replication-defective CpG-enriched vaccine strain IAV whose propagation is unimpaired in systems used for vaccine virus synthesis (MDCK cells and embryonated hens' eggs). This virus causes sub-clinical infection in mice that subsequently offers complete protection from challenge infection with a lethal dose of virus. CpG introduction is genetically stable. Altogether, the data presented herein support this approach for live attenuated vaccine augmentation.

We show a fundamental difference between human and chicken systems with respect to viral CpG sensing. While a homologue of ZAPL has been identified in chickens [22], no ZAPS homologue has been identified. That chicken ZAP may not have the same CpG binding specificity as human ZAP has previously been reported [21]. However, this is the first demonstration that a ZAP-sensitive CpG-enriched virus can be propagated equivalently to a WT virus in chicken systems. Possibly, chicken ZAP has lost CpG-specificity [21]. Although we showed in human-derived cells that ZAP is acting via transcriptional targeting and not via type I interferon signalling, it is possible that species-specific type I IFN signalling cascades may also

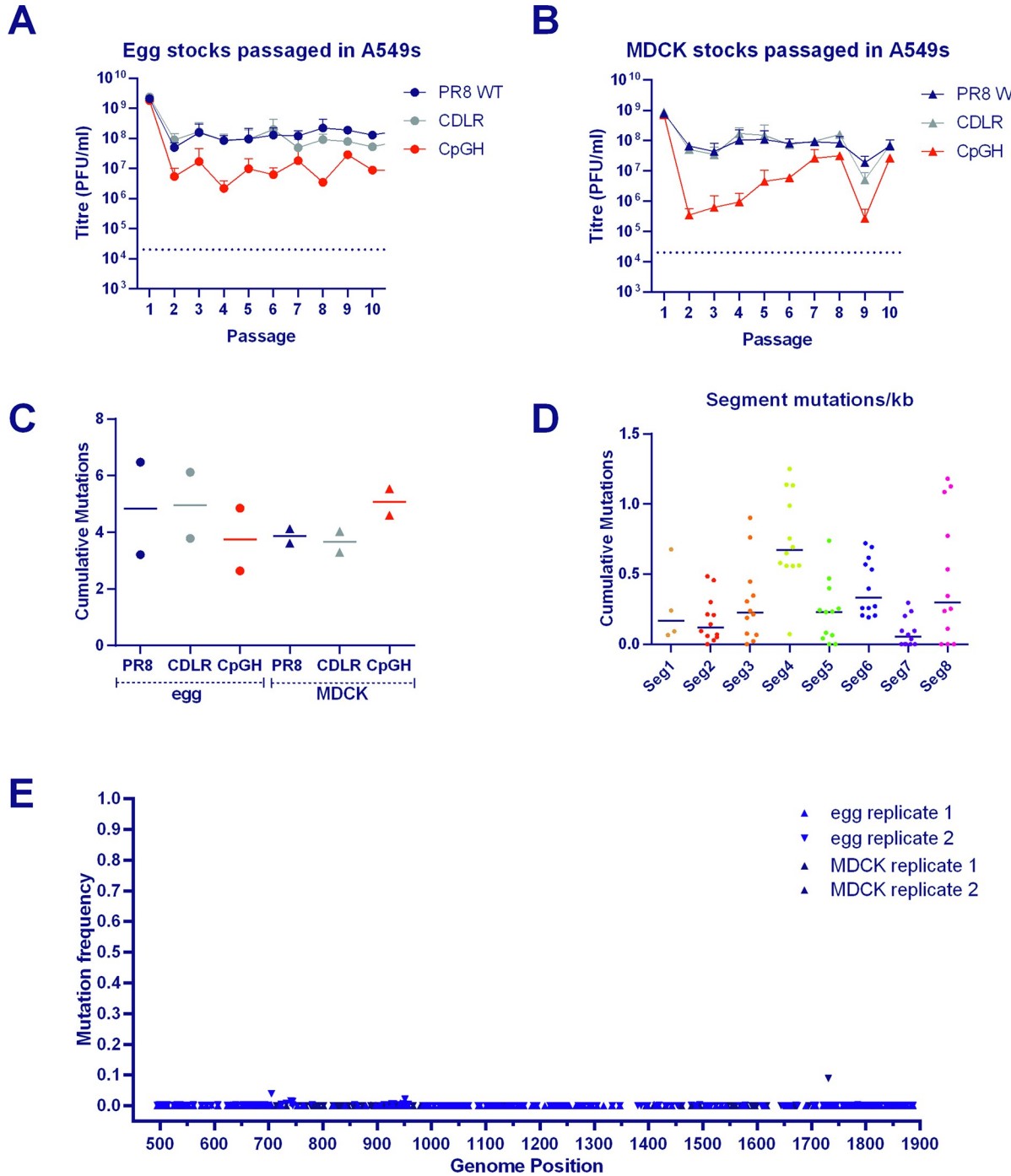

**Fig 8. Serial passage of CpG modified IAV does not enable deselection of CpG motifs.** Egg (**A**) or MDCK (**B**) derived virus stocks were passaged at an MOI of 0.01 ten times in A549 cells, and titred after each passage ($n = 2$). After ten passages, viruses were whole genome sequenced to test for reversion or epistatic compensatory mutations. (**C**). Total cumulative genome mutations after ten passages were calculated, with proportional representation of mutations that had not become fixed (e.g. if 10% of sequences contained a mutation at a given position, this scored as 0.1 mutations) calculated for egg virus stock passages and MDCK virus stock passages. (**D**). Cumulative mutations were calculated for each segment for each of the three viruses in the panel across four replicates (**E**). Mutation frequency at recoded positions across segment 1 for CpGH virus. The mutation frequencies of the four serially passaged replicates are overlaid. No mutations at CpG sites were observed. For sequence coverage, please refer to **S10 Fig** and for full genome mutational profiles to **S11 Fig**.

explain why chicken systems cannot target the CpGH virus. For example, chickens do not have RIG-I [66] and if CpG-specific responses signal via this pathway in chicken cells, chickens may have lost the ability to sense aberrant CpGs. Biochemical studies investigating RNA-protein interactions of ZAP and its cofactors in different species are needed in order to understand this further. Alternatively, PR8 may be highly effective at counteracting CpG sensing in chicken systems but less so in mammalian cells and in mice.

Understanding the mechanism by which CpG enrichment causes viral attenuation is paramount, as with large scale synonymous recoding it is possible that a mutant virus is only a few substitutions away from reversion to WT virulence. When mice were infected with a high dose of CpGH virus (**Fig 6A**), the initial response to infection was not dissimilar to that seen during WT virus infection. While the dose used was high, this highlights the importance of considering how CpGH-attenuated viruses may replicate in immumocompromised hosts. Furthermore, a key limitation of this approach is the possibility that an immunized population may vary in ZAP expression levels or functionality. The clinical importance of mutations in the ZAP-encoding gene, ZC3HAV1, was highlighted in a recent study that linked a mutation in this gene with COVID-19 disease severity [67]. For these reasons, multiple methods of viral attenuation integrated into a single platform should be considered; for example, the use of CpG enrichment to improve currently used live attenuated IAV vaccines that are replication-incompetent at 37°C [68–70].

Historically, use of synonymous recoding to attenuate IAV replication has focussed on disrupting codon pair biases [71–74]. In a more recent study that sought to uncouple the attenuative effects of codon pair bias deoptimisation from CpG enrichment in IAV, Groenke and co-authors found no attenuation of a CpGH IAV [75]. However, their CpGH IAV had 45 CpGs added, which was most comparable to our CpG-M mutant with 46 CpGs added and which was also not attenuated (**Fig 1F**) (though their experiments were undertaken using WSN strain recoded in segment 6, encoding the NA surface glycoprotein). Nevertheless, when disrupting codon pair biases, Groenke and colleagues maintained CpG frequencies in their recoded viruses, and found that use of disfavoured codon pairs attenuated IAV replication. Here, we have demonstrated the specificity of CpG-mediated attenuation through the restoration of CpGH virus fitness in ZAP-/- systems. Therefore, CpG enrichment and introduction of disfavoured codon pairs may represent discrete attenuation mechanisms.

CpG sensing by ZAP was shown to require introduction of only 15 CpGs for strong attenuation in a HIV-1 replicon [24]. In our IAV system, addition of 46 CpGs did not impart any attenuation, and adding 80 CpGs had a modest effect. ZAP was originally identified as a retrovirus restriction factor [9], and so possibly it more readily detects HIV-1 transcripts. Alternatively, a gene removed from HIV-1 pseudoviruses may counteract ZAP, IAV may more effectively neutralise ZAP than HIV-1, or ZAP sensing may be more active in the MT4 cells used for HIV-1 infections than the cell lines we used for IAV infections.

Interestingly, the anti-CpG effects of ZAP were mediated by transcript degradation and not by type I interferon signalling in the A549 cells used for infections. This lack of immune cascade contrasts with the antibody responses in mice to the CpGH virus that were equivalent to those elicited by WT virus. Further mechanistic understanding of how CpG-mediated attenuation occurs *in vitro* and *in vivo* is needed.

It has previously been reported that the IAV NS1 protein can specifically inhibit ZAPS or its binding partner TRIM25 [55,56] and that this can be ablated by mutation (Table 2). However, we were unable to exacerbate CpG-mediated attenuation by mutations in NS1 proposed to disrupt ZAP/TRIM25 targeting. Possibly, our assays were not sensitive enough to detect accentuation of a defect, or NS1 may target ZAP in this system. Nonetheless, the NS1 96/97

pair of mutations have previously been reported to lack specificity, likely due to NS1 misfolding [55,57].

CpG enrichment is increasingly being proposed as a strategy for live attenuated vaccine development [24,41,65]. We have shown that in a vaccine strain of IAV, CpG enrichment does not impair viral growth in embryonated hens' eggs or cultured MDCK cells–the main systems used for propagation of live attenuated IAV vaccines. This CpGH virus has desirable vaccine properties in mice, as it is clinically inapparent and highly protective from virus challenge. More widely, any live attenuated vaccines generated using a reverse genetics system and propagated in mammalian cells can be CpG-enriched and grown in ZAP-/- cells; therefore, large scale recoding is extrapolatable to other vaccine virus systems.

## Materials and methods

### Ethics statement

Mouse experiments were approved by the Roslin Institute Animal Welfare and Ethical Review Board under Home Office project license PF795AF60.

### Cells

Madin-Darby canine kidney (MDCK), human adenocarcinoma A549, human embryonic kidney (HEK)293T (all Sigma, Glasgow, UK), Normal Human Dermal Fibroblasts (NHDF; Thermo Fisher Scientific) and HEK293 cells and were grown in Dulbecco's Modified Eagle Medium (DMEM) (Sigma) supplemented with 10% v/v foetal calf serum (FCS) (Thermo Fisher Scientific) and 1% v/v penicillin/ streptomycin (Thermo Fisher Scientific). CLEC213 cells were grown in 1:1 Ham's F21 nutrient broth/ DMEM supplemented with 5% v/v FCS and 1% v/v penicillin/ streptomycin. Cells were maintained by twice-weekly passage. A549 ZAP -/- cells were a kind gift from the laboratory of Professor Sam Wilson [40], HEK293 TRIM25 -/- cells were a kind gift from the laboratory of Professor Gracjan Michlewski [42], and A549 KHNYN -/- cells were a kind gift from the laboratory of Dr Chad Swanson [19]. CLEC213 cells were a kind gift from Dr Sascha Trapp and generated by the laboratory of Dr Pascale Quéré [61]. Cells were checked for mycoplasma contamination regularly using a MycoAlert mycoplasma detection kit (Lonza, UK).

### Plasmids and plasmid mutagenesis

IAV reverse genetic plasmids corresponding with each of the 8 segments of IAV genome were a derivative of the UK National Institute of Biological Standards & Control A/Puerto Rico/8/ 1934 (PR8) vaccine strain [76]. For synonymous introduction of additional CpGs into segment 1 of the viral genome with compensatory mutations to maintain base frequencies ('CpGH') and UpA dinucleotide frequencies, constructs were designed using SSE software Simmonics v1.2 'mutate sequences' function [77]. The region of segment 1 targeted for mutation was selected to avoid previously characterised genome packaging signals and splice donor and acceptor sites for PB2-S1 [78]. A control mutant, 'CDLR', was also designed which was synonymously recoded over the same region, by reordering degenerate codons (e.g., the CUU and CUC codons for leucine may be switched within the coding sequence). The frequency of CpGs was not altered from the wildtype in this control. Using this approach, nucleotide, UpA dinucleotide and codon frequencies were also maintained. This control will test whether any previously unidentified RNA sequences or structures that are required for optimal virus replication have been unknowingly disrupted within the recoded region. A further control construct enriched for UpA dinucleotides ('UpAH'), was designed in the same way, with maintained

nucleotide, CpG dinucleotide and codon frequencies. Full segment sequences (**S1 Table**) are summarised. Codon pair and codon usage were calculated for the mutant panel using SSE v1.4 [77]. For recoding of a 1.42 kb region of segment 1, synthetic plasmid constructs were ordered from GeneArt (Thermo Fisher Scientific) flanked by BsmBI sites for cloning into pDUAL plasmids used for virus rescue, as previously described [76,79].

Mutagenesis of segment 8 was also performed to disrupt interactions of NS1 with CpG sensing pathways or various components of innate immune signalling; these are summarised (**Table 2**). Mutations were made on pDUAL plasmids encoding the PR8 gene [76] using a QuikChange II site directed mutagenesis kit (Agilent) according to manufacturer's instructions. Primers used are tabulated (**S2A Table**).

## Virus rescues

Viruses were rescued as previously described [59,80]. HEK293T cells in 6 well plates were transfected in Opti-MEM Reduced Serum Medium (Thermo Fisher Scientific) with 250 ng each of the 8 plasmids corresponding with each segment of viral genome (or 7 plasmids for mock) and 4 µl Lipofectamine 2000 transfection reagent (Thermo Fisher Scientific) per well. pDUAL plasmids were used for all segments, which are bi-directional; vRNA is under a polI promoter, and in the opposite orientation, positive sense RNA is under a polII promoter. After 24 hours, medium was replaced with serum free medium containing 1 µg/ml tosyl phenylalanyl chloromethyl ketone (TPCK)-treated trypsin (Sigma) and 0.14% BSA (w/v) (Sigma) (viral growth medium). At 48–72 hours post-transfection, supernatants were collected and passaged, either (1) onto MDCK cells in viral growth medium, with supernatants collected 48 hours later; or (2) 100 µl was inoculated into the allantoic fluid of an embyronated hen's egg at days 11–12 development. A packaging mutant, '4c6c' (multiple nucleotide changes in segment 4 and segment 6, encoding the haemagglutinin and neuraminidase viral surface proteins respectively) for use as a control for RNA:PFU ratio assays was propagated in MDCK cells rather than eggs to avoid reversion [36,46], as was an NS1 (segment 8) R38K41A mutant (defective in regulating interferon) [54].

## Egg infections

Dekalb White fertilised eggs were purchased from Henry Stewart (Norfolk, UK) and incubated at 37.5°C, 50% relative humidity until infection at day 11–12 (unless stated). Eggs were candled to confirm viability. Under sterile conditions, an egg punch was used to create two small holes in the shell–one in the air cavity and one in the locality of the allantoic fluid. 100 µl virus was injected into the allantoic fluid and both holes were sealed with sticky tape. For preparation of virus stocks, infected eggs were left for 48 hours prior to allantoic fluid harvest, which was spun down for 5 minutes at 800$g$ to pellet cellular debris, aliquoted and used as neat virus stock; presence of virus was confirmed by HA assay [81]. For comparison of viral titres between days 6 and 10, eggs were inoculated with 100 PFU of virus and were left for 24 hours prior to harvesting of allantoic fluid for virus titration. For kill curves, eggs were inoculated at day 10 with 0.001–1000 PFU virus and incubated until day 14 unless embryonic death was observed before that time. At experimental endpoints, eggs were chilled overnight and embryonic death was confirmed by membrane disruption.

## Virus titrations

Viral titres were quantified by plaque assays; 800 µl of ten-fold serial dilutions of viral stocks/ experiment supernatants/ allantoic fluid were incubated on sub-confluent MDCKs in 6-well plates for an hour, and then 2 ml overlay comprising 1:1 viral growth medium: 2.4% cellulose (Sigma) was added. Plates were incubated without disturbance for 48 hours, after which time 1

ml 10% formalin was added to each well and cells were fixed for at least 20 minutes. Overlays were removed and cells were stained with 0.1% toluidine blue (Sigma).

## Virus infections

For single cycle infections using a high multiplicity of infection (MOI), an MOI of 10 for 24 hours was used. For low multiplicity multi-cycle infections, an MOI of 0.01 was used in A549, 293 and MDCK cells. For NHDF infections, an MOI of 3 was used for 24 hour infections due to the lower permissiveness of these cells. Viruses were added to cells and incubated for 1 hour at 37˚C, and then inoculum was removed, cells were washed with serum free medium, and then either serum free medium (single cycle infection) or viral growth medium (multi-cycle infection) was added. Infected cells were incubated at 37˚C for the duration of the experiment.

## Generation of lentivirus vectors expressing ZAPL and ZAPS

In order to reconstitute isoform-specific ZAP expression in ZAP -/- A549 cells, lentiviruses expressing ZAPL or ZAPS isoforms under a polII promoter were produced using a pSCRPSY (accession number KT368137.1) vector-based system (a kind gift from the laboratory of Prof Sam Wilson). Both the A549 Cas9+ ZAP -/- cells and pSCRPSY carry puromycin resistance, and so a blasticidin S-resistance gene was cloned into the NheI site of pSCRPSY. pSCRPSY ZAPL and pSCRPSY ZAPS constructs were produced by cloning a cDNA of the open reading frame of ZAPL (accession number NM_020119.4) or ZAPS (accession number NM_024625.4) into the dual SfiI sites of pSCRPSY. For lentivirus production, the growth media on HEK293Ts at 80% confluency in a T175 flask was replaced with 25 ml of Opti-MEM. 6 μg PMD2.G (Addgene, 12259), 18 μg PsPAX2 (Addgene, 12260) and 12 μg of pSCRPSY (pSCRPSY, pSCRPSY-ZAPL or pSCRPSY-ZAPS) were added to 450 μl of Opti-MEM. 72 μl of Lipofectamine 2000 was added to 450 μl of Opti-MEM. Diluted plasmids and Lipofectamine were mixed and incubated for 30 minutes at room temperature. The transfection mixture was added to the flask and cells were then incubated at 37˚C, 5% $CO_2$ for 18h. Opti-MEM was then replaced with 25 ml of DMEM supplemented with 10% FBS. After a further 48 hours, supernatant containing P0 lentivirus was collected and centrifuged to remove cell debris. Due to low titre of ZAPL-expressing lentivirus, 75 ml of cell culture supernatant containing lentivirus was concentrated by ultracentrifugation using a SW32Ti rotor in a Beckman Coulter Optima Max-E ultracentrifuge at 50,000 $g$ for 2 hours. The viral pellet was then resuspended in 1 ml of 10% FCS-containing D-MEM. Lentivirus was aliquoted and stored at -80˚C.

## Generation of A549 cells expressing specific ZAP isoforms

A549 ZAP-/- cells at 70% confluency in a 6 well plate were transduced with ZAPL- or ZAPS-expressing pSCRPSY lentiviruses. 1 ml of lentivirus (pSCRPSY empty vector control, concentrated ZAPL, or ZAPS) or 10% FBS D-MEM as control, supplemented with polybrene (Merck, TR-1003-G) at 8 μg/ml, was added to each well of a 6 well plate. Spinoculation was performed (500 $g$, 1 hour). 48 hours after transduction, the inoculum was removed and cells were passaged in 10% FBS D-MEM supplemented with 50 μg/ml puromycin (Gibco) to select for transduced cells. Expression of the appropriate ZAP isoforms was confirmed by western blotting and confocal imaging (described below).

## Western blotting

Samples for western blotting were collected in 2X homemade Laemmli buffer [82], boiled for 10 minutes, cooled and loaded onto 10% pre-cast gels (Bio-Rad) for separation by SDS-PAGE.

Proteins were transferred onto nitrocellulose membranes (Thermo Fisher Scientific) using wet transfer (100V for 90 minutes on ice) and membranes were blocked using 5% milk (Morrisons, Edinburgh, UK) for at least 30 minutes. Membranes were washed three times with TBS-T (Tris-buffered-saline/0.1% Tween 20 (Sigma)) and incubated overnight at 4°C with primary antibodies in 2% BSA/TBS-T (0.1%). Commercial antibodies were used for β-tubulin (clone YL1/2, Bio-Rad, Watford, UK; 1:5000), β-actin (Abcam ab8227; 1:2000), ZAP (Invitrogen PA5-31650; 1:1000), TRIM25 (Abcam ab167154; 1:1000) and KHNYN (Insight Bio sc-514168; 1:1000). In-house rabbit polyclonal antibodies raised against NP and PB2 are previously described [83] and were both diluted 1:1000. Membranes were then washed 3 times and incubated with species-specific secondary antibodies conjugated to Alexafluor-680 or -800 (Thermo Fisher Scientific) diluted 1:5000 in 2% BSA/TBS-T (0.1%) for 90 minutes. Membranes were washed 3 times and visualised using a LICOR Odyssey Fc imaging system.

## Confocal imaging

Immunofluorescence was used to confirm ZAP expression in lentivirus-transduced A549 cells, and presence of viral NP in chicken enteroids. For imaging of A549 cells, A549 pSCRPSY-transduced cells at 95% confluency on coverslips in 24 well plates were fixed using 500 μl of 3.7% neutral buffered formalin (CellPath) for 15 minutes, and then washed with PBS. Cells were permeabilised using 500 μl of 0.02% Triton X100 for 5 minutes and blocked for 30 minutes in 2% FCS. Anti-ZC3HAV1 antibody (16820-1-AP) used 1:500 was diluted in 2% FCS/PBS and incubated on cells overnight at 4°C. 200 μl of 1:1000 AlexaFluor-488 anti-rabbit secondary antibody (Thermo Fisher Scientific, A32731) was added for 1 hour. Cell nuclei were stained using 1:5000 Hoechst (Thermo Fisher Scientific) in PBS for 10 minutes. Between addition of each reagent, cells were washed 3 times using 500 μl of PBS. All images were acquired with Zeiss LSM 710 confocal microscope fitted with 40X/1.4 oil-immersion objective lens. Controls for background signal were mock infected cells and infected cells stained using secondary antibodies only.

For organoid imaging, 50 organoids in a 24 well plate were fixed in 4% neutral buffered formalin for 45 minutes and washed with PBS. Organoids were then permeabilised and blocked with 5% v/v goat serum (Sigma) in permeabilisation solution (0.5% v/v bovine serum albumin and 0.1% v/v Saponin (Merck, Glasgow, UK) in PBS) for 30 minutes. Antibodies were diluted in permeabilisation solution and incubated with organoids (mouse monoclonal NP antibody AA5H clone (Abcam, Cambridge, UK ab20343) used 1:500 overnight at 4°C, phalloidin-647 (Thermo Fisher A22287) used 1:50 and incubated for 1 hour at room temperature). 200 μl of 1:1000 AlexaFluor-488 anti-mouse secondary antibody (Thermo Fisher A32723) was added for 1 hour. Cell nuclei were stained using 1:5000 Hoechst in PBS for 10 minutes. Between addition of each reagent, cells were washed 3x using 500 μl of PBS. All images were acquired with Zeiss LSM 880 confocal microscope fitted with 40X/1.4 oil-immersion objective lens. Secondary-only and isotype (mouse IgG2b) controls were used.

## Reverse transcription (RT)-quantitative (q)PCR

RT-qPCR was performed as previously described [79]. RNA was extracted from cells using Qiagen QiAmp Viral RNA mini kits. Nucleic acid extracts were treated using RQ1 RNAse-free DNAse (Promega) according to manufacturer's instructions. qPCR was performed in technical triplicates using BioLine (London, UK) SensiFAST One Step RT-qPCR Lo-Rox kit (BIO-82020), with primers targeting a conserved region of segment 1 or segment 5 of IAV PR8 strain (S2B Table). Cycling conditions were 45°C for 10 minutes, then 40 cycles of 95°C for 10

seconds then 60˚C for 30 seconds. Melt curves were performed between 50˚C-99˚C with one degree increments at second intervals.

### Urea-PAGE RNA gels

IAV virions equivalent to $10^{10}$ PFU from egg stocks were semi-purified by centrifugation through a buffered 25% sucrose cushion (w/v) (100 mM NaCl, 10 mM Tris-HCl [pH 7], 1 mM EDTA) using a SW32Ti rotor in a Beckman Coulter Optima Max-E ultracentrifuge at 280,000 *g* for 90 minutes at 4˚C. RNAs were extracted from virions using an RNEasy mini extraction kit (Qiagen) following resuspension of the pellets directly into 350 μl of buffer RLT. Extracted RNA was separated by PAGE using homemade 5% urea polyacrylamide gels in 1X Tris-borate-EDTA (TBE) buffer (89mM Tris-borate, 2mM EDTA [pH 8.3]) run for 6 hours at 120 V. Viral gene segments were visualised by silver staining using the Silver Stain Plus Kit (Bio-Rad) according to manufacturer's instructions and imaged using a Samsung Xpress C480FW scanner.

### Bisulphite conversion

RNA was extracted from ~$5x10^8$ IAV virions (egg virus stock) using a Qiagen RNEasy mini extraction kit, followed by Promega RQ1 DNAse treatment. Bisulphite conversion reactions were performed on extracted egg stock RNAs or *in vitro* generated genomic RNA (gRNA; negative sense) transcripts using the Zymo (Freiburg, Germany) RNA bisulphite conversion kit (EZ RNA Methylation Kit, RS001). In the reaction, cytosines are converted to uracils unless protected by 5-methylcytosine ($m^5C$) methylation. Following conversion, RNA libraries for sequencing were prepared using the Takara (Saint-Germain-en-Laye, France) low input RNA-seq kit and sequenced on a MiSeq Micro v2 150PE flowcell at Edinburgh Genomics (Edinburgh UK). Sequencing data handling and analyses were performed using the Galaxy platform [84]. Adapter sequences (AGATCGGAAGAGCACACGTCTGAACTCCAGTCA and AGATCGGAAGAGCGTCGTGTAGGGAAAGAGTGT) were trimmed from reads using cutadapt and output sequences were joined using fastq-join. Sequences were aligned to an *in silico* converted PR8 segment 1 (where all cytosines in the negative sense were replaced with thymines) using Bowtie2 [85] and resultant BAM datasets were converted to tabular pileup format to allow for calculation of the proportion of retained cytosines (indicating presence of 5-mc).

Specific regions selected for further analysis were amplified by RT-PCR from the converted RNA. RNAs were reverse transcribed using SuperScript III (Invitrogen) with random hexamer primers according to manufacturer's instructions. PCRs were performed using specific primers (**S2C Table**) and Q5 High-Fidelity Polymerase (NEB, Hitchin, UK) according to manufacturer's instructions. Cycling conditions were 98˚C for 30 seconds, then 45 cycles of 98˚C for 10 seconds, 58˚C for 21 seconds then 72˚C for 30 seconds with a final extension of 72˚C for 2 minutes. Amplicons were sent for sequencing using the Amplicon-EZ service (Genewiz, Leipzig, Germany). Sequences were analysed as above except that adapter trimming was performed using the primer sequences.

### HEK Blue assay

To assay type I IFN production, A549 cells in 24 well plates were infected at an MOI of 10 as described above but in the absence of trypsin. At 10 hours post infection, supernatants were harvested and viruses were inactivated by exposure to 120 mJ/cm$^2$ of UV for 10 minutes in a UVP CL-1000 UV crosslinker. 20 μl of inactivated sample or type I IFN standard (Abcam, 500, 50 or 5 pg/ μl) was added to HEK-Blue IFNα/β cells (InvivoGen hkb-ifnab) at $4x10^4$ cells/ well

in 96 well format. HEK-Blue cells have a gene for secreted alkaline phosphatase incorporated under an ISG54 promoter (that is robustly induced in the presence of type I IFN). Cells were incubated with supernatant from A549 infections for 24 hours. HEK-Blue cell supernatants were then collected and added to QuantiBlue reagent (InvivoGen) according to manufacturer's instructions. After 15–30 minutes, when a colour change was visible in the positive control (titrated IFN), absorbance was read at 620 nm, as a correlate of IFN concentration.

## Minigenome assays

Minigenome assays were performed as previously described [59]. Culture medium on 70–90% confluent HEK293T cells in 24 well plates was replaced with Opti-MEM. Cells were then transfected with 50 ng/well (unless stated) of plasmids encoding IAV segments 1, 2, 3, and 5 (sufficient for viral ribonucleoprotein (vRNP) formation) with positive orientation under a PolII promoter (the same pDUAL plasmid constructs used for virus rescue were used here), along with 20 ng/ well of a reporter construct expressing an IAV-like vRNA with a luciferase coding sequence in place of an IAV open reading frame. At 48 hours post-transfection, cells were harvested in 100 μl 5X reporter lysis buffer (Promega, Chilworth, UK) and refrigerated to allow cell debris to settle to the bottom of the tube. 60 μl supernatant was added to 25 μl 600 μM beetle luciferin (Promega) in a white walled 96 well plate, and luminescence was measured on a GloMax luminometer (Promega). Each data point was represented by four technical repeats, and each assay was undertaken for at least four biological repeats.

## *In vitro* transcription assays

PCR amplicons were generated for the production of positive or negative sense, full-length segment 1 RNA by the addition of a T7 promoter sequence to the forward or reverse primer, respectively (**S2D Table**). PCRs were performed using Q5 DNA polymerase kit (NEB) with the appropriate primer pair and 25 ng of plasmid as template (segment 1 of PR8, CDLR, or CpGH). Cycling conditions were 95˚C for 5 minutes, then 30 cycles of 95˚C for 30 seconds, 50˚C for 30 seconds then 72˚C for 2 minutes, with a final extension of 72˚C for 5 minutes. Amplicons were purified by agarose gel extraction using the MinElute Gel Extraction kit (Qiagen) and quantified using the Qubit dsDNA BR Assay kit (Thermo Fisher). RNA transcripts were generated with 40 ng of these amplicons as template using the MEGAscript T7 Transcription kit (Thermo Fisher) in half reaction volumes according to manufacturer's instructions for the indicated times. Following incubation, transcript concentrations were determined using the Qubit RNA HS Assay kit (Thermo Fisher).

## *In vitro* translation assays

*In vitro* generated RNA transcripts above were used templates for *in vitro* translation assays using a Rabbit Reticulocyte Lysate kit (Promega) at the indicated concentrations. Reactions were performed at 1/5th volume with the inclusion of Transcend Biotin-Lysyl-tRNA (Promega) to allow antibody independent detection of all translated products. Samples were assayed by western blot (as above) with the exception that membranes were blocked with 5% BSA/TBS for 1 hour, incubated with IRDye 800CW Streptavidin (LICOR) for 1 hour, washed 3 times and visualised using the LICOR Odyssey Fc imaging system.

## Northern blotting

RNA probes and positive control full-length, negative sense transcripts were generated by first producing PCR amplicons of the target sequence with the addition of a T7 promoter sequence

to the forward or reverse primer, (S2E Table). PCRs were performed using Q5 DNA polymerase kit (NEB) with the appropriate primer pair and 25 ng of plasmid as template (Seg1-PR8 or Seg5-PR8). Cycling conditions were 95˚C for 5 minutes, then 30 cycles of 95˚C for 30 seconds, 50˚C for 30 seconds then 72˚C for 2 minutes, with a final extension of 72˚C for 5 minutes. Amplicons were purified by agarose gel extraction using the MinElute Gel Extraction kit (Qiagen) and quantified using the Qubit dsDNA BR Assay kit. RNA transcripts were generated with 40 ng of these amplicons as template using the MEGAscript T7 Transcription kit (Thermo Fisher) in half reaction volumes for 4 hours according to manufacturer's instructions except for the inclusion of Biotin-11-UTP (Cambridge BioScience) at a molar ratio of 2:3 with unlabelled UTP for the generation of labelled probes. Following incubation, transcript concentrations were determined using the Qubit RNA HS Assay kit (Thermo Fisher Scientific).

Parental A549 Cas9+ or A549 ZAP-/- cells were infected at a high MOI as described above and cell pellets were collected at 8 hours post infection. RNA was extracted from cell pellets using the RNeasy Mini Kit (Qiagen). RNA gels were run, blotted and hybridised using reagents from the NorthernMax–Gly Kit (Thermo Fisher Scientific) unless otherwise indicated. Briefly, RNA samples were run on 1% Agarose-LE gels at low voltage for 4 hours along with 1 ng each of positive control positive sense segment 1 and segment 5 transcripts and RiboRuler High Range RNA ladder (Thermo Fisher Scientific). Prior to transfer, gels were and scanned using a LICOR Odyssey Fc imaging system to visualise 18s rRNA. Nucleic acids were blotted on to Nytran SuPerCharge nylon membranes (Cytiva) by overnight downward transfer using the Turbo Blotter System apparatus (Cytiva). Membranes were baked at 68˚C for 10 minutes and crosslinked by exposure to 1200 (x100) J/cm$^2$ twice in CL-1000 UV crosslinker. Membranes were pre-hybridised in ULTRAhyb Ultrasensitive Hybridisation Buffer for 30 minutes at 68˚C and hybridised overnight at 68˚C with biotinylated probes at a concentration of 0.1 nM. Membranes were washed once in 2x SCC, 0.1% SDS at room temperature for 15 minutes and three times with 0.1x SCC, 0.1% SDS at 68˚C for 15 minutes each. Membranes were blocked in 5% dried milk powder in TBS-T for 4 hours at room temperature, washed six times with TBS-T and incubated with HRP-conjugated Streptavidin (Proteintech) in 5%BSA/TBS-T for 30 minutes. Membranes were washed four times in TBS-T, incubated briefly with LumiGLO Peroxidase Chemiluminescent Substrate (Insight Bio) and visualised by exposure to Amersham High Performance Chemiluminescence film (Cytiva) for 5 minutes.

## Mouse infections

Five-week-old female BALB/c mice, maintained in groups of six, were purchased from Harlan (Oxon, UK) and housed for one week prior to infections. Prior to infection, mice were weighed to ensure that there were no differences in mean weights or deviations across groups. For infection experiments, mice were inoculated intranasally with 200 PFU virus (either PR8, CDLR, CpGH or mock) under brief isofluorane anaesthesia. Mice were weighed daily until day 5, when mice were euthanised by terminal isofluorane anaesthesia. Groups were blinded during weighing. Left lungs were harvested in 2 ml Eppendorf (Stevenage, UK) Safe-Lock micro test tubes containing 500 μl DMEM and a stainless-steel carbide bead (Qiagen), and snap frozen on dry ice until virus titration. For titration, lungs were thawed on wet ice and fragmented using a Qiagen TissueLyser II at 200 oscillations/minute for 2 minutes or until tissue was visibly homogenised. Suspended tissue was then serially diluted and titrated by plaque assay. For challenge infections, groups of ten six-week old mice were inoculated with 20 PFU virus (either PR8, CDLR, CpGH or mock) under brief isofluorane anaesthesia, except for the PR8 group which comprised six mice. Mice were weighed and monitored for clinical signs until 10 days post-infection. At 20 days post-infection, tail bleeds were performed. At 21 days,

5 mice from each group (or 6 from PR8) were culled by terminal isofluorane anaesthesia and cardiac puncture. Sera were collected. The remaining mice were challenged with 200 PFU PR8 WT virus (due to licensing constraints inoculation then challenge with the same PR8 WT virus was not possible). Mice were monitored until cull at 5 days post-challenge by terminal isofluorane anaesthesia and cardiac puncture, with sera and left lung collected as described above for ELISA and plaque assays respectively.

## Mouse serum ELISA

To test mouse sera cross-reactivity with IAV virions, virions were semi-purified by overlaying 5 mL of clarified PR8, CDLR or CpGH virus stocks on 25% buffered sucrose (w/v) cushion (100 mM NaCl, 10 mM Tris-HCl pH7, 1 mM EDTA) in a Beckman Coulter Optima Max-E ultracentrifuge with a SW32Ti rotor at 60,000 $g$ for 2 hours at 4˚C. Viral pellets were resuspended in 500 μL PBS. Virions equivalent to an input of $2 \times 10^6$ PFU were diluted in 100 μL of 0.1 M carbonate-bicarbonate buffer (pH 9.6) (Sigma-Aldrich) and immobilised on 96-well ELISA plates (Greiner Bio-One) overnight at 4˚C. Plates were washed twice with PBS-Tween 20 (0.05% v/v) and blocked with 1% BSA/PBS at room temperature for 2 hours. Mouse sera diluted 1/100 in 1% BSA/PBS were then applied as primary antibody along with a 5-fold dilution series of a positive control serum designated as 100 arbitrary units (AU)/μl and plates were incubated at room temperature with agitation for 2 hours. Plates were washed six times with PBS-T and incubated with HRP-conjugated goat anti-mouse IgG (H+L) (1:2000) (Bio-Rad) in 1% BSA/PBS for 1 hour with agitation. Plates were washed thrice with PBS-T and incubated with 2,2'-azino-bis(3'-ethylbenzothiazoline-6-sulfonic acid) substrate (SLS) after which the colorimetric change was arrested by adding 70 μL/well 1% SDS. Optical density was read at 405 nm on a Cytation 3 Cell Imaging Multi-Mode Reader (Agilent) and analysed in Bioek Gen5 (Agilent).

## Dot blotting with mouse sera

To test mouse sera cross-reactivity with IAV proteins, MDCK cells were infected at MOI of 10 with PR8 strain for 8 hours, and infected cell lysates were harvested in 2X Laemmli buffer. A single lysate long well was run out by SDS-PAGE and transferred to a nitrocellulose membrane as described above. Membranes were placed in a strip-blot and individual lanes were incubated with day 20 sera from individual mice diluted 1:100 in 2% BSA/PBS for 1 hour with agitation. Western blotting then proceeded as described above.

## Chicken enteroid infections

Chicken enteroids were prepared as described previously [63]. Briefly, small intestinal tissue was removed from Hy-Line Brown embryos supplied by the National Avian Research Facility, Edinburgh, UK (Home Office Project Licenses PE263A4FA) and cut into 5 mm sections. Tissue was digested and villi were isolated and cultured. At day 2 of culture, approximately 200 enteroids were infected with $10^5$ virus particles/ enteroid for 1 hour, washed 3 times in PBS and then incubated for 48 hours in the presence of 2 μg/mL TPCK trypsin for virus quantification by plaque assay, confocal imaging, or arrayed gene expression analyses.

## Fluidigm analysis of innate immune response gene expression in infected chicken enteroids

At 48 hpi, enteroids were suspended in RNALater and RNA was isolated using Tru-Seq total RNA Sample Preparation v2 kit (Illumina, Great Abingdon, UK) according to manufacturer's

instructions. cDNA was synthesized from 100 ng of RNA using High Capacity Reverse Transcription kit (Life Technologies, UK) according to manufacturer's instructions with a random hexamer primer and oligo(dT). cDNA was pre-amplified as previously described [86] and analysed through quantitative PCR (qPCR) with the microfluidic 96.96 Dynamic array (Fluidigm, San Francisco, CA) performed on the BioMark HD instrument (BioMark, Boise, ID). Raw Cq values were processed with GenEx6 and GenEx Enterprise (MultiD Analyses AB), with correction for primer efficiency and reference gene normalisation. Reference genes used were TATA box binding protein (TBP), Tubulin alpha chain (TUBA8B), beta-actin (ACTB), beta-glucuronidase (GUSB), glyceraldehyde-3-phosphate dehydrogenase (GAPDH) and ribosomal 28S (r28S).

### Serial passage

Viruses were serially passaged ten times at an MOI of 0.01 in A549 cells. Viruses were titred by plaque assay after each passage. Biological duplicates were performed with viruses rescued from both eggs and from MDCK cells. At the end of ten passages, viruses were sequenced. RNA was extracted from culture supernatants using Viral RNA mini kits (Qiagen) according to manufacturer's instructions. Extracted RNAs were reverse transcribed using SuperScript III (Invitrogen) with the IAV vRNA specific Uni12 primer (AGCAAAAGCAGG) [87] according to manufacturer's instructions. PCR primers were designed to amplify approximately 400 base fragments with approximately 60 base overlaps across the genome (**S2F Table**). Additional primers were designed specific to the segment 1 and segment 5 recoded regions as appropriate. Four sets of primer pools each containing 10 non-overlapping sets of primer pairs were generated (1A-3E odd, 1B-3F even, 3G-8B odd and 4A-8C even) with each primer at a final concentration of 5 μM. PCRs were performed using specific primers and Q5 High-Fidelity Polymerase according to manufacturer's instructions with primers at a concentration of 500 nM each in the final reactions. Cycling conditions were 98°C for 30 seconds, then 45 cycles of 98°C for 20 seconds, 55°C for 20 seconds then 72°C for 30 seconds with a final extension of 72°C for 2 minutes. The 4 multiplexed amplicon sets for each sample were purified using a PureLink PCR Purification kit, quantified using a Qubit dsDNA BR assay kit (Invitrogen) and pooled at equimolar ratios. Amplicon pools were sent for sequencing using the Amplicon-EZ service (Genewiz). Sequencing data handling and analyses were performed using the Galaxy platform [84]. Primer sequences were trimmed from reads using cutadapt and output sequences were joined using fastq-join. Sequences were aligned to the PR8 reference genome or genome sequences with recoded segment 1, as appropriate, using Bowtie2 [85]. Variants and coverage levels in the resultant BAM datasets were analysed using iVar [88] variants with tabular outputs.

### Statistical analyses and data presentation

For all *in vitro* experiments, at least 3 biological repeats with 3 technical repeats were performed. Individual data points represent independent biological repeats, calculated from a mean of two or three technical repeats, and horizontal bars represent the mean. To assess for statistical differences between multiple groups, one way ANOVAs were performed for multiple comparisons. For comparisons across two groups, ratio paired t tests were performed in GraphPad Prism v9 unless data are presented as ratios, in which case paired t tests were performed.

### Supporting information

**S1 Table. Segment 1 modified PR8 sequences.**
(DOCX)

**S2 Table. Primers used for mutagenesis and sequencing. S2A Table.** Primers used for site directed mutagenesis of IAV PR8 segment 8. **S2B Table.** Primers for amplification of influenza A virus A/Puerto Rico/8/1934 (PR8) viral transcripts by qPCR. **S2C Table.** Primers for amplification of specific bisulphite converted RNA regions. **S2D Table.** Primers for generation of *in vitro* transcription amplicon templates. **S2E Table.** Primers for Northern blotting probe amplicon templates. **S2F Table.** Primers for overlapping full genome sequencing.
(DOCX)

**S1 Fig. Validation of ZAP pathway inhibition assays. A.** ZAP -/- A549 cells (37) were validated by western blotting. **B.** TRIM25 -/- 293 cells (38) were validated by western blotting. **C.** KHNYN -/- A549 cells (18) were validated by western blotting. **D.** Distribution of CpGs in segment 1 of recoded viruses. **E.** Distance between CpG sites in recoded region of segment 1. **F.** Codon pair scores across the virus mutant panel. **G.** Relative synonymous codon usage (RSCU) across the virus mutant panel.
(TIF)

**S2 Fig. Validation of ZAP expression reconstitution in ZAP-/- A549 cells delivered by lentivirus vector and examined by confocal imaging.**
(TIF)

**S3 Fig. Bisulphite conversion does not identify any conserved m$^5$C methylation sites in the IAV genome, and CpG enrichment does not introduce any such methylation events.** Methylation signal identified across the genome of wildtype PR8 (**A**), CDLR (**B**) and CpGH (**C**). While the segment 1 sequence was different in A-C, the genome sequence across segments 2–8 was identical, and so for these segments, the data are representative of 3 biological repeats.
(TIF)

**S4 Fig. Minireplicon assays confirmed decreased transcript and protein production associated with CpG enrichment of PR8 segment 1.** Minireplicon assays that reconstitute the viral polymerase were performed using a luciferase reporter (**A**, luciferase reporter signal; **B**, protein signal by western blot).
(TIF)

**S5 Fig. Sera from mice infected with 20 PFU of CpGH virus contain antibodies that are IAV-specific.** 6-week old female BALB/c mice were infected with wildtype PR8 (n = 6), CDLR (n = 10) or CpGH (n = 10) viruses. After 20 days, tail bleeds were performed and sera harvested from all mice. **A.** For mice yielding sufficient sera, serum was diluted 1:100 in 2% BSA/ PBS and cross-reacted with cell lysate from MDCK cells infected with wildtype PR8 virus at MOI of 10 for 8 hours. Whole anti-IAV antibody (αH1N1 USSR77) was used as a positive control. Variable cross-reactivity to HA and NP proteins was observed for the different virus inocula, but this variability was consistent across viruses (**B**).
(TIF)

**S6 Fig. Plaque phenotype of synonymously recoded viruses.** WT PR8, CDLR and CpGH viruses were rescued in embryonated hens' eggs and titred by plaque assay in MDCK cells. Representative images taken from plaque assays performed at the same time.
(TIF)

**S7 Fig. Survival of chicken embryos infected at day 10 post fertilisation.** Embryonated hens' eggs at developmental day 10 were inoculated with ten-fold serial dilutions of virus stock (**A,** PR8; **B,** CDLR; **C,** CpGH) and survival was monitored by candling every 12 hours for 96

hours. The number of eggs inoculated at each dose is indicated in brackets.
(TIF)

**S8 Fig. Confocal imaging of infected chicken enteroids.** WT PR8, CDLR and CpGH viruses were used to infect chicken enteroids at $10^5$ PFU/enteroid for 48 hours, and viral protein production was assessed visually using confocal microscopy. Representative z-axis projections of chicken enteroid whole-mount stained to detect cell nuclei (Hoechst, blue), F-actin-expressing brush border (red) and virus nucleoprotein (green). Scale bar: 20 μm.
(TIF)

**S9 Fig. Innate immune gene expression levels of WT PR8, CDLR and CpGH viruses in chicken organoids.**
(TIF)

**S10 Fig. Sequence coverage depth for deep sequenced serially passaged viruses recoded in segment 1, over the complete genome. A-D**, PR8 wildtype virus; **E-H**, CDLR; **I-L**, CpGH; **M**, input plasmid.
(TIF)

**S11 Fig. Full genome mutation traces for all serially passaged viruses. A-D**, PR8 wildtype virus; **E-H**, CDLR; **I-L**, CpGH; **M,** input plasmid.
(TIF)

# Acknowledgments

We are grateful to the labs of Prof Sam Wilson (MRC-University of Glasgow Centre for Virus Research, UK) for provision of A549 ZAP-/- cell lines and pSCRPSY lentivirus vector, Prof Gracjan Michlewski (University of Edinburgh, UK/ International Institute of Molecular and Cell Biology, Poland) for the provision of TRIM25-/- HEK293 cells, Dr Chad Swanson (King's College London, UK) for A549 KHNYN-/- cells and Dr Sascha Trapp (French National Institute for Agriculture, Food and Environment, France) for the provision of CLEC213 cells. We are indebted to the staff in the Roslin Biological Research Facility for technical support for mouse infection experiments and mouse line maintenance, to the staff of the Central Services Unit, Facilities team, Bioimaging facilities, and to the National Avian Research Facility for supply of eggs in the Roslin Institute, University of Edinburgh.

We are grateful to Christina Vrettou (The Roslin Institute, UK) for technical assistance, to Dr James Glover (The Roslin Institute, UK) for critical feedback on the manuscript and to Dr Edward Hutchinson (MRC-University of Glasgow Centre for Virus Research, UK) for helpful discussions.

# Author Contributions

**Conceptualization:** Colin P. Sharp, Beth H. Thompson, Helen Wise, Sara Clohisey Hendry, Finn Grey, Lonneke Vervelde, Peter Simmonds, Paul Digard, Eleanor R. Gaunt.

**Data curation:** Colin P. Sharp, Eleanor R. Gaunt.

**Formal analysis:** Colin P. Sharp, Eleanor R. Gaunt.

**Funding acquisition:** Peter Simmonds, Eleanor R. Gaunt.

**Investigation:** Colin P. Sharp, Beth H. Thompson, Tessa J. Nash, Ola Diebold, Rute M. Pinto, Luke Thorley, Samantha Sives, Helen Wise, Eleanor R. Gaunt.

**Methodology:** Colin P. Sharp, Beth H. Thompson, Tessa J. Nash, Rute M. Pinto, Luke Thorley, Yao-Tang Lin, Samantha Sives, Sara Clohisey Hendry, Finn Grey, Lonneke Vervelde, Peter Simmonds, Paul Digard, Eleanor R. Gaunt.

**Project administration:** Sara Clohisey Hendry, Eleanor R. Gaunt.

**Resources:** Finn Grey, Lonneke Vervelde, Peter Simmonds, Paul Digard.

**Software:** Peter Simmonds.

**Supervision:** Colin P. Sharp, Lonneke Vervelde, Paul Digard, Eleanor R. Gaunt.

**Validation:** Colin P. Sharp, Beth H. Thompson, Tessa J. Nash, Eleanor R. Gaunt.

**Visualization:** Colin P. Sharp, Beth H. Thompson, Tessa J. Nash, Eleanor R. Gaunt.

**Writing – original draft:** Colin P. Sharp, Lonneke Vervelde, Paul Digard, Eleanor R. Gaunt.

**Writing – review & editing:** Colin P. Sharp, Ola Diebold, Rute M. Pinto, Yao-Tang Lin, Samantha Sives, Finn Grey, Peter Simmonds, Paul Digard, Eleanor R. Gaunt.

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
