## [Decision Letter · Decision Letter 0]

3 Jan 2023

Dear Dr Gaunt,

Thank you very much for submitting your manuscript "CpG dinucleotide enrichment in the influenza A virus genome as a live attenuated vaccine development strategy" for consideration at PLOS Pathogens. As with all papers reviewed by the journal, your manuscript was reviewed by members of the editorial board and by several independent reviewers. In light of the reviews (below this email), we would like to invite the resubmission of a significantly-revised version that takes into account the reviewers' comments.

Your manuscript was sent out for review and while the work was viewed as a significant addition to the field there were concerns raised by the reviewers that need to be addressed. In particular reviewer #2 raised several important points that need to be addressed to interpret and/or validate the experiments. First of all a description of the statistical analysis utilized throughout the manuscript and an explanation as to the number of times an experiment was performed needs to be included in order to interpret much of the data. In addition, although the animal experiments appear to have 6-10 mice per group, as written, it appears these experiments were only an N of 1. These studies need to be repeated at least 1 time, possibly more depending upon the results and statistical analysis. In addition, reviewer 2 raises the need for additional control viruses as outlined in his review. Both reviewers have also identified areas that need additional discussion of previously published work or findings from your work that warrant further discussion.

We cannot make any decision about publication until we have seen the revised manuscript and your response to the reviewers' comments. Your revised manuscript is also likely to be sent to reviewers for further evaluation.

Sincerely,

Deborah Lenschow

Academic Editor

PLOS Pathogens

Benhur Lee

Section Editor

PLOS Pathogens

Kasturi Haldar

Editor-in-Chief

PLOS Pathogens

orcid.org/0000-0001-5065-158X

Michael Malim

Editor-in-Chief

PLOS Pathogens

orcid.org/0000-0002-7699-2064

Your manuscript was sent out for review and while the work was viewed as a significant addition to the field there were concerns raised by the reviewers that need to be addressed. In particular reviewer #2 raised several important points that need to be addressed to interpret and/or validate the experiments. First of all a description of the statistical analysis utilized throughout the manuscript and an explanation as to the number of times an experiment was performed needs to be included in order to interpret much of the data. In addition, although the animal experiments appear to have 6-10 mice per group, as written, it appears these experiments were only an N of 1. These studies need to be repeated at least 1 time, possibly more depending upon the results and statistical analysis. In addition, reviewer 2 raises the need for additional control viruses as outlined in his review. Both reviewers have also identified areas that need additional discussion of previously published work or findings from your work that warrant further discussion.

Reviewer's Responses to Questions

**Part I - Summary**

Reviewer #1: This paper reports the construction and properties of a variant PR8 influenza A virus with elevated CpG in segment 1. This is a nice extension of recent similar work from the Bieniasz lab to study the effects of CpG addition to virus properties in another virus context (e.g. PMID: 36075961, and less importantly, PMID: 33402534). As has recently been shown with EV-71, viruses can be attenuated in ZAP-positive cells and not in ZAP KO cells. Analysis of viral RNA in CpG-high IFA virus here showed a reduction in segment 1 RNA levels, and also other segments, in ZAP-positive cells. The CpGhigh virus at low dose inoculation was able to induce antibodies in mice, and protect from subsequent challenge with wt virus, while only causing mild symptoms.

The work here shows also that the virus can be propagated to high titer in chicken eggs and cell lines, which are apparently not expressing significant ZAP function, and thus is suitable for large-scale vaccine preparation. Passage of the CpGhigh virus in restrictive A549 cells did not readily result in reversion or loss of the CpG mutations.

I consider this paper to be a significant addition to the literature describing attempts to generate attenuated viruses for vaccine utilization. It is clearly only an extension of the ground-breaking paper from the Bieniasz lab on similar work with EV71, but this is a very important setting (influenza virus). All the work seems solid. The impact of the heavy mutagenesis on replication is perhaps surprisingly small (30x or so in Zap positive cells—and there is a lot of noise) but apparently sufficient to render the virus poorly able to cause symptoms on its own while still inducing protective immunity.

I think a limitation of this whole approach – the dependence of the immunized population on variable ZAP functionality – and the potential for patient variability in their levels of ZAP expression and functionality – needs to be discussed. Essential!

Reviewer #2: Sharp and colleagues increase the number of CpG dinucleotides in segment 1 of influenza A virus strain Puerto Rico (PR8). They record reduced virus titers of the recoded virus (CPGH) on A549 cells and, to a lower extent, on 293 cells. Using ZAP-negative A549 and TRIM25-negative 293 cells, and using siRNA treatment the authors conclude that reduced virus replication is mediated via the ZAP-TRIM25 axis. In contrast, CpG-rich IAV replicate similar to wild-type viruses or a recoded control virus with similar efficiency on MDCK cells and in embryonated eggs. Further, the authors show that packaging of viral RNAs is not affected by CpG enrichment and that the methylation profile remains unaffected.

The authors claim that the CPGH mutant virus is a promising candidate for generation of an attenuated live vaccine because the virus induces protective immunity in a mouse model of infection and the virus exhibits genetic stability after serial passage in cultured cells.

The strengths of the manuscript are that the experimental procedures were performed and are presented in a logical and straightforward fashion. Weaknesses include that similar experiments (i.e., an increase of CpG dinucleotides) were done and reported before. Quite a few of the conclusions are hard to follow as details are missing (see below).

**Part II – Major Issues: Key Experiments Required for Acceptance**

Reviewer #1: I have no major issues requiring additional experimentation. There are some writing issues.

The review of the history and background of ZAP is in a few places slightly off base.

Lines 48 ff: Actually there are even more spliced forms (see MacDonald papers, PMID: 31118263). Might as well mention this. Probably all the forms (including S and L) bind CpG, since this is due to the N-terminal finger region shared by them all. Doesn’t really make sense to suggest that “a consensus mechanism for CpG sensing by ZAP remains unresolved”. The structure of the RNA-protein complex is well-established and has been reported (eg. PMID: 31719195).

The IFN inducibility of ZAPS, and the constitutive expression of ZAPL, and their relative activity has been an issue in the field but the data in Supplementary Fig 1 are striking and very clear and significantly add to the literature. It should be noted that this is in a particular cell line.

Lines 113 ff: TRIM25 is not actually “required” but stimulates ZAP activity. The findings in NHDF cells may be due to incomplete KD as suggested, but may also simply be due to lesser need for TRIM25 in these cells. The authors should cite more comprehensive screens for cofactors, which have revealed numerous stimulatory factors (as many as 31 (PMID: 22615998)).

Line 145: The increase in replication of all viruses, including CpG-high ones, with ZAPL expression is curious and needs some discussion. And the defect in replication of all viruses with ZAPS expression, is also odd, though at least the CpG-high virus was more affected than the CpG low virus. To some extent these finding undercut the arguments of the paper.

Line 182: Wouldn’t deamination of C give U, not T, in RNA? The switch to T only occurs in DNA, and only after replication of the deaminated U to A, followed by replication of the A to give T in the next round. I’m thinking this is major faux pas. Correct me if I’m wrong, or fix the text.

Line 214: There is a failed logic here. Yes, ZAPS is an ISG (an IFN stimulated gene), but this is not necessarily relevant to its IFN-stimulating activity. These two issues need to be separated and discussed as such.

While the number of CpGs are tested (Figure 1F and Supplementary Fig S1D), it has been shown that the spacing and clustering of the CpGs is very important (Bi PMID: 36075961). Some discussion of these issues is needed for the particular design of segment 1 used here, as related to the Bieniasz paper.

Surely this important paper should be cited?

The role of ZAP and OAS3/RNAseL pathways in the attenuation of an RNA virus with elevated frequencies of CpG and UpA dinucleotides. Odon V, Fros JJ, Goonawardane N, Dietrich I, Ibrahim A, Alshaikhahmed K, Nguyen D, Simmonds P. Nucleic Acids Res. 2019 Sep 5;47(15):8061-8083. doi: 10.1093/nar/gkz581. PMID: 31276592

Also:

CpG-Recoding in Zika Virus Genome Causes Host-Age-Dependent Attenuation of Infection With Protection Against Lethal Heterologous Challenge in Mice.

Trus I, Udenze D, Berube N, Wheler C, Martel MJ, Gerdts V, Karniychuk U. Front Immunol. 2020 Jan 24;10:3077. doi: 10.3389/fimmu.2019.03077. eCollection 2019. PMID: 32038625

Reviewer #2: 1) Nowhere in the manuscript is any reference to the statistical analyses that were conducted to be found. Hence, this reviewer is unable to accept or reject many of the conclusions made herein.

2) The scrambled/control virus CDLR where "codon rearrangement" was performed needs to a) be explained better and b) complemented by viruses in which, for example, UpA's are enriched and/or codon/bicodon frequencies are altered.

3) The animal experiments, apparently, were not done in a randomized, blinded fashion? In addition, since the mice were group-housed as I understand from the methods described, one could argue that the groups have an N=1. As the CpGH virus seemed not to be attenuated until day 3 after inoculation, there should be at least one repeat of the experiment.

**Part III – Minor Issues: Editorial and Data Presentation Modifications**

Reviewer #1: See above.

Reviewer #2: 1) Line 108 and Figure 1: The term "Low MOI" has to be clarified (.1, .01?). I also assume that the dots represent independent replicates and that the horizontal bar mean/median? Why were foreskin fibroblasts used?

There is reference to the "significant" reduction in virus titers when wt/CDLR is compared to CpGH. However, there is substantial variability (~20-fold) within wt. Hence, growth kinetics rather than a single time point (why 48 h) would seem appropriate.

2) Lines 178 -218 and Figure 3: How do you explain that methylation patterns of the CpGH virus remain unaltered? was that the case for the other CpG mutant viruses (5', 3') as well? If there is no difference, is there a need to show the data?

3) Figure 4: Again, the differences in virus titers are quite variable between replicates. In addition, the conclusion that N and P2 protein levels are down in CpGH viruses is not supported by the data as presented (Fig. 4C).

4) Figure 5: PB2 and N seem to be expressed at lower levels in the case of the CDLR virus (Fig. 5C). Why?

Also, there is lower levels of vRNA but not =RNA in the case of CDLR. Can you explain?

5) Figure 6 and Lines 280-306/667-680: The animal experiments needs a better design/repeat and proper statistical evaluation (blinded investigators, randomization, exclusion of cage effect). Also, how do you explain the parallel curves of all three viruses until day 3 (Fig. 6A) and the recovery of the CpGH mice? How would the CpGH virus do in immunosuppressed mice, which would be important to know if this virus were to be considered as a live attenuated vaccine?

6) Figure 7: Again, on MDCK, titers stretch over two orders of magnitude. Again, growth kinetics would seem more appropriate than just one time point.

PLOS authors have the option to publish the peer review history of their article (what does this mean?). If published, this will include your full peer review and any attached files.

Reviewer #1: No

Reviewer #2: No
---

## [Decision Letter · Decision Letter 1]

30 Mar 2023

Dear Dr Gaunt,

Thank you very much for submitting your manuscript "CpG dinucleotide enrichment in the influenza A virus genome as a live attenuated vaccine development strategy" for consideration at PLOS Pathogens. As with all papers reviewed by the journal, your manuscript was reviewed by members of the editorial board and by several independent reviewers. The reviewers appreciated the attention to an important topic. Based on the reviews, we are likely to accept this manuscript for publication, providing that you modify the manuscript according to the review recommendations.

The authors both agree that this is an important piece of work and that the majority of the concerns raised during the first review have been addressed adequately by the authors. However, reviewer #2 raises an important point regarding the need to place this work in the context of published literature in the field, in particular to calculate the codon and bicodon usage/scores and to discuss the findings of your work in the context of what exists in the literature and the contradictions/controversy that exists.

Sincerely,

Deborah Lenschow

Academic Editor

PLOS Pathogens

Benhur Lee

Section Editor

PLOS Pathogens

Kasturi Haldar

Editor-in-Chief

PLOS Pathogens

orcid.org/0000-0001-5065-158X

Michael Malim

Editor-in-Chief

PLOS Pathogens

orcid.org/0000-0002-7699-2064

The authors both agree that this is an important piece of work and that the majority of the concerns raised during the first review have been addressed adequately by the authors. However, reviewer #1 raises an important point regarding the need to place this work in the context of published literature in the field, in particular to calculate the codon and bicodon usage/scores and to discuss the findings of your work in the context of what exists in the literature and the contradictions/controversy that exists.

Reviewer Comments (if any, and for reference):

Reviewer's Responses to Questions

**Part I - Summary**

Reviewer #1: As before, this paper presents an important extension of previous work on the use of CpG-enriched viral genomes a immunogens to influenza. I feel the paper will be significant.

Reviewer #2: The authors have addressed most of my comments. Importantly, the authors have included the description of the statistical analyses and one of the mutant viruses (UpAH), which behaved as predicted: It has a clear growth defect, which, in contrast to the CpGH virus, cannot be rescued on ZAP-/- cells. However, this result obviously raises the question what the mechanism of reduced growth of this virus is.

The authors raise the issue of codon usage, which was unaltered in the recoded viruses. However, given the controversies between the "Simmonds" and the "Wimmer/Skiena/Mueller/Coleman" groups on the reason for recoding, it would seem necessary to a) calculate not only codon but also bicodon usage/scores and b) at least discuss the findings in the context of the literature abound on recoding of flu viruses. There is not a single reference referring to this earlier work. Here is a (certainly not comprehensive) selection of relevant literature:

1: Groenke N, Trimpert J, Merz S, Conradie AM, Wyler E, Zhang H, Hazapis OG,

Rausch S, Landthaler M, Osterrieder N, Kunec D. Mechanism of Virus Attenuation

by Codon Pair Deoptimization. Cell Rep. 2020 Apr 28;31(4):107586. doi:

10.1016/j.celrep.2020.107586. PMID: 32348767.

2: Kaplan BS, Souza CK, Gauger PC, Stauft CB, Robert Coleman J, Mueller S,

Vincent AL. Vaccination of pigs with a codon-pair bias de-optimized live

attenuated influenza vaccine protects from homologous challenge. Vaccine. 2018

Feb 14;36(8):1101-1107. doi: 10.1016/j.vaccine.2018.01.027. PMID: 29366707.

3: Broadbent AJ, Santos CP, Anafu A, Wimmer E, Mueller S, Subbarao K. Evaluation

of the attenuation, immunogenicity, and efficacy of a live virus vaccine

generated by codon-pair bias de-optimization of the 2009 pandemic H1N1 influenza

virus, in ferrets. Vaccine. 2016 Jan 20;34(4):563-570. doi:

10.1016/j.vaccine.2015.11.054. Epub 2015 Dec 2. PMID: 26655630; PMCID:

PMC4713281.

4: Mueller S, Coleman JR, Papamichail D, Ward CB, Nimnual A, Futcher B, Skiena

S, Wimmer E. Live attenuated influenza virus vaccines by computer-aided rational

design. Nat Biotechnol. 2010 Jul;28(7):723-6. doi: 10.1038/nbt.1636. Epub 2010

Jun 13. PMID: 20543832; PMCID: PMC2902615.

5: Wimmer E, Mueller S, Tumpey TM, Taubenberger JK. Synthetic viruses: a new

opportunity to understand and prevent viral disease. Nat Biotechnol. 2009

Dec;27(12):1163-72. doi: 10.1038/nbt.1593. PMID: 20010599; PMCID: PMC2819212.

Of particular interest seems reference 1, in which high CpG content of a recoded influenza A virus seems not to have caused any growth defect in cultured cells including A549. Is the difference observed here to published data a segment-specific phenomenon? Is it a strain-specific phenomenon? At any rate, I think the results presented here need to be discussed in the light of earlier and at least partially contradictory results.

**Part II – Major Issues: Key Experiments Required for Acceptance**

Reviewer #1: I think the responses to our other reviewer have addressed the statistical limitations of the initial draft.

Reviewer #2: (No Response)

**Part III – Minor Issues: Editorial and Data Presentation Modifications**

Reviewer #1: I think the minimal concession about the potential issues of patient variability in ZAP functionality should be extended -- this could be a serious problem in the application of these viruses in vaccines.

Reviewer #2: (No Response)

PLOS authors have the option to publish the peer review history of their article (what does this mean?). If published, this will include your full peer review and any attached files.

Reviewer #1: No

Reviewer #2: No

Figure Files:

Data Requirements:

Reproducibility:

References:

---

## [Editor Report · Decision Letter 2]

12 Apr 2023

Dear Dr Gaunt,

We are pleased to inform you that your manuscript 'CpG dinucleotide enrichment in the influenza A virus genome as a live attenuated vaccine development strategy' has been provisionally accepted for publication in PLOS Pathogens.

Best regards,

Deborah Lenschow

Academic Editor

PLOS Pathogens

Benhur Lee

Section Editor

PLOS Pathogens

Kasturi Haldar

Editor-in-Chief

PLOS Pathogens

orcid.org/0000-0001-5065-158X

Michael Malim

Editor-in-Chief

PLOS Pathogens

orcid.org/0000-0002-7699-2064

The authors have modified this version of the manuscript to put their work in the context of already published literature and they have included the data requested by the reviewers. As noted before, both reviewers felt this was an important piece of work for the field.
---

## [Editor Report · Acceptance letter]

2 May 2023

Dear Dr Gaunt,

We are delighted to inform you that your manuscript, "CpG dinucleotide enrichment in the influenza A virus genome as a live attenuated vaccine development strategy," has been formally accepted for publication in PLOS Pathogens.

Best regards,

Kasturi Haldar

Editor-in-Chief

PLOS Pathogens

orcid.org/0000-0001-5065-158X

Michael Malim

Editor-in-Chief

PLOS Pathogens

orcid.org/0000-0002-7699-2064